# No Pairs Left Behind: Improving Metric Learning with Regularized Triplet Objective

## Abstract

We propose a novel formulation of the triplet objective function that improves metric learning without additional sample mining or overhead costs. Our approach aims to explicitly regularize the distance between the positive and negative samples in a triplet with respect to the anchor-negative distance. As an initial validation, we show that our method (called *No Pairs Left Behind* [NPLB]) improves upon the traditional and current state-of-the-art triplet objective formulations on standard benchmark datasets. To show the effectiveness and potentials of NPLB on real-world complex data, we evaluate our approach on a large-scale healthcare dataset (UK Biobank), demonstrating that the embeddings learned by our model significantly outperform all other current representations on tested downstream tasks. Additionally, we provide a new model-agnostic single-time health risk definition that, when used in tandem with the learned representations, achieves the most accurate prediction of subjects' future health complications. Our results indicate that NPLB is a simple, yet effective framework for improving existing deep metric learning models, showcasing the potential implications of metric learning in more complex applications, especially in the biological and healthcare domains. Our code package as well as tutorial notebooks is available on our public repository: <revealed after the double blind reviews>.

## 1 Introduction

Metric learning is the task of encoding similarity-based embeddings where similar samples are mapped closer in space and dissimilar ones afar (Xing et al., 2002; Wang et al., 2019; Roth et al., 2020). Deep metric learning (DML) has shown success in many domains, including computer vision (Hermans et al., 2017; Vinyals et al., 2016; Wang et al., 2018b) and natural language processing (Reimers & Gurevych, 2019; Mueller & Thyagarajan, 2016; Benajiba et al., 2019). Many DML models utilize paired samples to learn useful embeddings based on distance comparisons. The most common architectures among these techniques are the Siamese (Bromley et al., 1993) and triplet networks (Hoffer & Ailon, 2015). The main components of these models are the: (1) Strategies for constructing training tuples and (2) objectives that the model must minimize. Though many studies have focused on improving sampling strategies (Wu et al., 2017; Ge, 2018; Shrivastava et al., 2016; Kalantidis et al., 2020; Zhu et al., 2021), modifying the objective function has attracted less attention. Given that learning representations with triplets very often yield better results than pairs using the same network (Hoffer & Ailon, 2015; Balntas et al., 2016), our work focuses on improving triplet-based DML through a simple yet effective modification of the traditional objective.

Modifying DML loss functions often requires mining additional samples or identifying new quantities (e.g. identifying class centers iteratively throughout training (He et al., 2018)) or computing quantities with costly overheads (Balntas et al., 2016), which may limit their applications. In this work, we aim to provide an easy and intuitive modification of the traditional triplet loss that is motivated by metric learning on more complex datasets, and the notion of density and uniformity of each class. Our proposed variation of the triplet loss leverages all pairwise distances between existing pairs in traditional triplets (positive, negative, and anchor) to encourage denser clusters and better separability between classes. This allows for improving already existing triplet-based DML architectures using implementations in standard deep learning (DL) libraries (e.g. TensorFlow), enabling a wider usage of the methods and improvements presented in this work.

Many ML algorithms are developed for and tested on datasets such as MNIST (LeCun, 1998) or ImageNet (Deng et al., 2009), which often lack the intricacies and nuances of data in other fields, such as health-related domains (Lee & Yoon, 2017). Unfortunately, this can have direct consequences when we try to understand how ML can help improve care for patients (e.g. diagnosis or prognosis). In this work, we demonstrate that DML algorithms can be effective in learning embeddings from complex healthcare datasets. We provide a novel DML objective function and show that our model's learned embeddings improve downstream tasks, such as classifying subjects and *predicting future health risk using a single-time point*. More specifically, we build upon the DML-learned embeddings to formulate a new mathematical definition for patient health-risks using a single time point which, to the best of our knowledge, does not currently exist. To show the effectiveness of our model and health risk definition, we evaluate our methodology on a large-scale complex public dataset, the UK Biobank (UKB) (Bycroft et al., 2018), demonstrating the implications of our work for both healthcare and the ML community. In summary, our most important contributions can be described as follows. 1) We **present a novel triplet objective function** that improves model learning without any additional sample mining or overhead computational costs. 2) We **demonstrate the effectiveness of our approach on a large-scale complex public dataset** (UK Biobank) *and* on conventional benchmarking datasets (MNIST, Fashion MNIST (Xiao et al., 2017) and CIFAR10 (Krizhevsky, 2010)). This demonstrates the potential of DML in other domains which traditionally may have been less considered. 3) We **provide a novel definition of patient health risk from a single time point**, demonstrating the real-world impact of our approach by predicting *current healthy subjects'* future risks using only a single lab visit, a challenging but crucial task in healthcare.

## 2  BACKGROUND AND RELATED WORK

Contrastive learning aims to minimize the distance between two samples if they belong to the same class (are similar). As a result, contrastive models require two samples to be inputted before calculating the loss and updating their parameters. This can be thought of as passing two samples to two parallel models with tied weights, hence being called *Siamese* or *Twin* networks (Bromley et al., 1993). Triplet networks (Hoffer & Ailon, 2015) build upon this idea to rank positive and negative samples based on an anchor value, thus requiring the model to produce mappings for all three before the optimization step (hence being called triplets).

**Modification of Triplet Loss:** Due to their success and importance, triplet networks have attracted increasing attention in recent years. Though the majority of proposed improvements focus on the sampling and selection of the triplets, some studies (Balntas et al., 2016; Zhao et al., 2019; Kim & Park, 2021; Nguyen et al., 2022) have proposed modifications of the traditional triplet loss proposed in Hoffer & Ailon (2015). Similar to our work, Multi-level Distance Regularization (MDR) (Kim & Park, 2021) seeks to regularize the DML loss function. MDR regularizes the pairwise distances between embedding vectors into multiple levels based on their similarity. The goal of MDR is to disturb the optimization of the pairwise distances among examples and to discourage positive pairs from getting too close and the negative pairs from being too distant. A drawback of regularization methods is the choice of hyperparameter that balances the regularization term, though adaptive balancing methods could be used (Chen et al., 2018; Heydari et al., 2019). Most related to our work, Balntas et al. (2016) modified the traditional objective by explicitly accounting for the distance between the positive and negative pairs (which the traditional triplet function does not consider), and applied their model to learn local feature descriptors using shallow convolutional neural networks. They introduce the idea of "in-triplet hard negative", referring to the swap of the anchor and positive sample if the positive sample is closer to the negative sample than the anchor, thus improving on the performance of traditional triplet networks (we refer to this approach as *Distance Swap*). Though this method uses the distance between the positive and negative samples to choose the anchor, it does not explicitly enforce the model to regularize the distance between the two, which was the main issue with the original formulation. Our work addresses this pitfall by using the notion of local density and uniformity (defined later in §3) to explicitly enforce the regularization of the distance between the positive and negative pairs using the distance between the anchors and the negatives. As a result, our approach ensures better inter-class separability while encouraging denser intra-class embeddings. In addition to MDR and Swap Distance, we benchmark our approach againt three related and widely-used metric learning algorithms, namely *LiftedStruct* Song et al. (2015), *N-Pair Loss Sohn (2016)*, and InfoNCE Oord et al. (2018a). Due to the space constraints, and given the popularity of the methods, we provide an overview of these algorithms in Appendix E

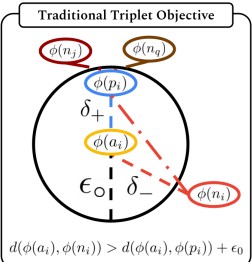 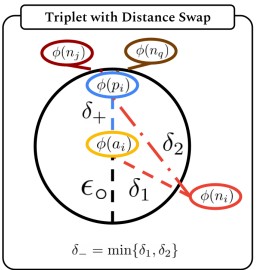 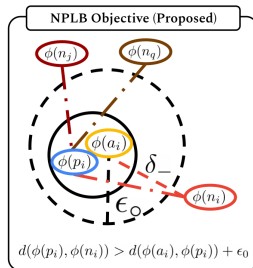

Figure 1: **Visual comparisons between a traditional triplet loss (left), a Distance Swap triplet loss (middle) and our proposed No Pairs Left Behind objective (right) on a toy example.** In this figure $\phi$ refers to a learned operator, $\delta_+ = d(\phi(p_i), \phi(a_i))$, $\delta_- = d(\phi(n_i), \phi(a_i))$ and $\epsilon_\circ$ denotes the margin. For this toy example, the network $\phi(\cdot)$ trained on the traditional objective (left) is only concerned with satisfying $\delta_- > \delta_+ + \epsilon_\circ$, potentially mapping a positive close to negative samples $n_j, n_q$, which is not desirable. A similar case could happen for the Distance Swap variant as well (middle). Our proposed objective seeks to avoid this by explicitly forming dependence between the distance of the positive and negative pair and $\delta_-$. This regularization results in denser mappings when samples are similar and vice versa when the samples are dissimilar, as shown in §4. We describe our formulation and modification in §3.

**Deep Learned Embeddings for Healthcare:** Recent years have seen an increase in the number of DL models for Electronic Health Records (EHR) with several methods aiming to produce rich embeddings to better represent patients (Rajkomar et al., 2018; Choi et al., 2016b; Tran et al., 2015; Nguyen et al., 2017; Choi et al., 2016a; Pham et al., 2017). Though most studies in this area consider temporal components, DeepPatient (Miotto et al., 2016) does not explicitly account for time, making it an appropriate model for comparison with our representation learning approach given our goal of predicting patients' health risks using a single snapshot. DeepPatient is an unsupervised DL model that seeks to learn general deep representations by employing three stacks of denoising autoencoders that learn hierarchical regularities and dependencies through reconstructing a masked input of EHR features. We hypothesize that learning patient reconstructions alone (even with masking features) does not help to discriminate against patients based on their similarities. We aim to address this by employing a deep metric learning approach that learns similarity-based embeddings.

**Predicting Patient's Future Health Risks:** Assessing patients' health risk using EHR remains a crucial, yet challenging task of epidemiology and public health (Li et al., 2015). An example of such challenges are the *clinically-silent* conditions, where patients fall within "normal" or "borderline" ranges for specific known blood work markers, while being at the risk of developing chronic conditions and co-morbidities that will reduce quality of life and cause mortality later on (Li et al., 2015). Therefore, early and accurate assessment of health risk can tremendously improve the patient care, specially in those who may appear "healthy" and do not show severe symptoms. Current approaches for assessing future health complications tie the definition of health risks to multiple time points (Hirooka et al., 2021; Chowdhury & Tomal, 2022; Razavian et al., 2016; Kamal et al., 2020; Cohen et al., 2021; Che et al., 2017). Despite the obvious appeal of such approaches, the use of many visits for modeling and defining risk simply ignores a large portion of patients who do not return for subsequent check ups, especially those with lower incomes and those without adequate access to healthcare (Kullgren et al., 2010; Taani et al., 2020; Nishi et al., 2019). Given the importance of addressing these issues, we propose a mathematical definition (that is built upon DML) based on a single time point, which can be used to predict patient health risk from a *single lab visit*.

## 3 METHODS

**Main Idea of *No Pairs Left Behind* (NPLB):** The main idea behind our approach is to ensure that, during optimization, the distance between positive $p_i$ and negative samples $n_i$ is considered, and regularized with respect to the anchors $a_i$ (i.e. explicitly introducing a notion of distance between $d(p_i, n_i)$ which depends on $d(a_i, n_i)$). We visualize this idea in Fig. 1. The mathematical intuition behind our approach can be described by considering in-class *local density* and *uniformity*, as introduced in Rojas-Thomas & Santos (2021) for unsupervised clustering evaluation metric.

Given a metric learning model $\phi$, let local density of a sample $p_i$ be defined as $LD(p_i)_{p_i \in c_k} = \min\{d(\phi(p_i), \phi(p_j))\}, \forall p_i \in c_k$ and $i \neq j$, and let $AD(c_k)$ be the average local density of all point in class $c_k$. An ideal operator $\phi$ would produce embeddings that are compact while well

separated from other classes, or that the in-class embeddings are *uniform*. This notion of uniformity, is proportional to the difference between the local and average density of each class, i.e.

$$
Unif(c_k) = \begin{cases} \frac{\sum_i^{|c_k|} |LD(p_i) - AD(c_k)|}{AD(c_k) + \xi} & \text{if } |c_k| > 1 \\ 0 & \text{Otherwise} \end{cases}.
$$

for $0 < \xi \ll 1$. However, computing density and uniformity of classes is only possible post-hoc once all labels are present and not feasible during training if the triplets are mined in a self-supervised manner. To reduce the complexity and allow for general use, we utilize *proxies* for the mentioned quantities to regularize the triplet objective using the notion of uniformity. We take the distance between positive and negative pairs as inversely proportional to the local density of a class. Similarly, the distance between anchors and negative pairs is closely related to the average density, given that a triplet model maps positive pairs inside an $\epsilon_\circ$-ball of the anchor ($\epsilon_\circ$ being the margin). In this sense, the uniformity of a class is inversely proportional to $|d(\phi(p_i), \phi(n_i)) - d(\phi(a_i), \phi(n_i))|$.

**NPLB Objective**: Let $\phi(\cdot)$ denote an operator and $T$ be the set of triplets of the form $(p_i, a_i, n_i)$ (positive, anchor and negative tensors) sampled from a mini-batch $B$ with size $N$. For the ease of notation, we will write $\phi(q_i)$ as $\phi_q$. Given a margin $\epsilon_\circ$ (a hyperparameter), the traditional objective function for a triplet network is shown in Eq. (1):

$$
\mathcal{L}_{Triplet} = \frac{1}{N} \sum_{(p_i, a_i, n_i) \in T}^{N} [d(\phi_a, \phi_p) - d(\phi_a, \phi_n) + \epsilon_\circ]^+ \tag{1}
$$

with $[\cdot]^+ = \max\{\cdot, 0\}$ and $d(\cdot)$ being the Euclidean distance. Minimizing Eq. (1) only ensures that the negative pairs fall outside of an $\epsilon_\circ$-ball around the $a_i$, while bringing the positive sample $p_i$ inside of this ball (illustrated in Fig. 1), satisfying $d(\phi_a, \phi_n) > d(\phi_a, \phi_p) + \epsilon_\circ$. However, this objective does not explicitly account for the distance between positive and negative samples, which can impede performance especially when there exists high in-class variability. Motivated by our main idea of having denser and more uniform in-class embeddings, we add a simple regularization term to address the issues described above, as shown in Eq. (2)

$$
\mathcal{L}_{NPLB} = \frac{1}{N} \sum_{(p_i, a_i, n_i) \in T}^{N} [d(\phi_a, \phi_p) - d(\phi_a, \phi_n) + \epsilon_\circ]^+ + [d(\phi_p, \phi_n) - d(\phi_a, \phi_n)]^p, \tag{2}
$$

where $p \in \mathbb{N}$ and $NPLB$ refers to "No Pairs Left Behind." The regularization term in Eq. (2) enforces positive and negative samples to be roughly the same distance away as all other negative pairings, while still minimizing their distance to the anchor values. However, if not careful, this approach could result in the model learning to map $n_i$ such that $d(\phi_a, \phi_p) > \max\{\epsilon_\circ, d(\phi_p, \phi_n)\}$, which would ignore the triplet term, resulting in a minimization problem with no lower bound[1]. To avert such issues, we restrict $p = 2$ (or generally, $p \equiv 0 \pmod 2$) as in Eq. (3).

$$
\mathcal{L}_{NPLB} = \frac{1}{N} \sum_{(p_i, a_i, n_i) \in T}^{N} [d(\phi_a, \phi_p) - d(\phi_a, \phi_n) + \epsilon_\circ]^+ + [d(\phi_p, \phi_n) - d(\phi_a, \phi_n)]^2, \tag{3}
$$

Note that this formulation does not require mining of any additional samples nor complex computations since it just uses the existing samples in order to regularize the embedded space. Moreover

$$
\mathcal{L}_{NPLB} = 0 \implies -[d(\phi_p, \phi_n) - d(\phi_a, \phi_n)]^2 = [d(\phi_a, \phi_p) - d(\phi_a, \phi_n) + \epsilon_\circ]^+
$$

which, considering only the real domain, is possible if and only if $d(\phi_p, \phi_n) = d(\phi_a, \phi_n)$, and $d(\phi_a, \phi_n) \geq d(\phi_a, \phi_p) + \epsilon_\circ$, explicitly enforcing separation between negative and positive pairs.

---

[1] The mentioned pitfall can be realized by taking $p = 1$, i.e.

$$
\mathcal{L}(p_i, a_i, n_i) = \frac{1}{N} \sum_{(p_i, a_i, n_i) \in T}^{N} [d(\phi_a, \phi_p) - d(\phi_a, \phi_n)) + \epsilon_\circ]^+ + [d(\phi_p, \phi_n) - d(\phi_a, \phi_n)].
$$

In this case, the model can learn to map $n_i$ and $a_i$ such that $d(\phi_a, \phi_n) > C$ where $C = \max\{d(\phi_p, \phi_n), d(\phi_a, \phi_p) + m\}$, resulting in $\mathcal{L} < 0$.

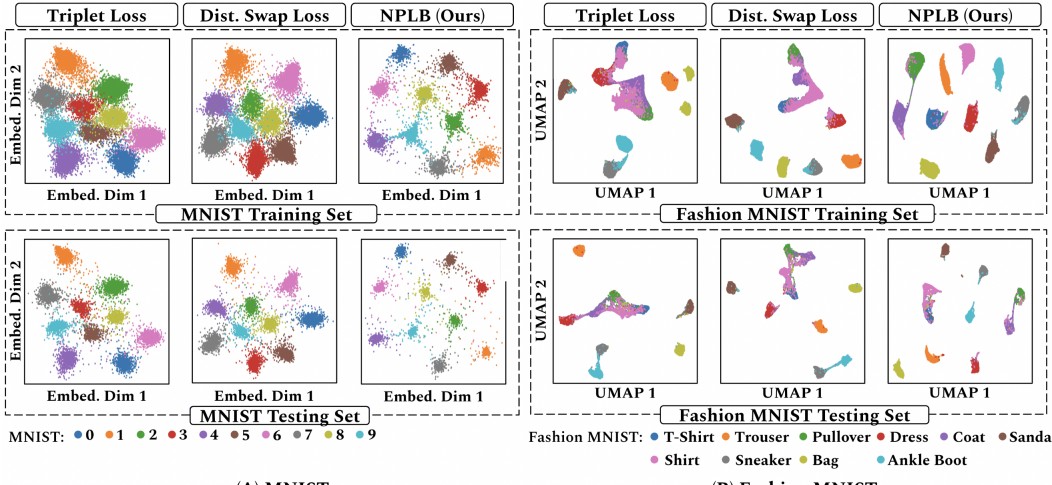

Figure 2: **Visual comparisons between a traditional triplet loss, Distance Swap and the proposed NPLB objective on train (top row) and test (bottom row) sets of (A) MNIST and (B) Fashion MNIST.** *(A)* To evaluate the feasibility of the proposed Triplet loss on general datasets, we trained the same network (described in Appendix Appendix K) under identical conditions on the MNIST dataset, with the only difference being the loss function used. *(B)* UMAP-reduced embeddings of Fashion MNIST trained with the three triplet loss versions. Our results indicate both quantitative and qualitative improvements in the embeddings, as shown above and in Table 1. (See Fig. A6 and A7 for higher resolution versions).

## 4    VALIDATION OF *NPLB* ON STANDARD DATASETS

Prior to testing our methodology on healthcare data, we validate our derivations and intuition on common benchmark datasets, namely MNIST, Fashion MNIST and CIFAR10. To assess the improvement gains from the proposed objective, we refrained from using more advanced triplet construction techniques and followed the most common approach of constructing triplets using the labels offline. We utilized the same architecture and training settings for all experiments, with the only difference per dataset being the objective functions (see Appendix K for details on each architecture). After training, we evaluated our approach quantitatively through assessing classification accuracy of embeddings produced by optimizing the traditional triplet, Swap Distance and our proposed NPLB objective. The results for each data are presented in Table 1, showing that our approach improves classification. We also assessed the embeddings qualitatively: Given the simplicity of MNIST, we designed our model to produce two-dimensional embeddings which we directly visualized. For Fashion MNIST and CIFAR, we generated embeddings in $\mathbb{R}^{64}$ and $\mathbb{R}^{128}$, respectively, and used Uniform Manifold Approximation and Projection (UMAP) (McInnes et al., 2018) to reduce the dimensions for visualizations, as shown in Fig. 2. Our results show that networks trained on NPLB objective produce embeddings that are denser and well separated in space, as desired.

Table 1: **Comparison of state-of-the-art (SOTA) triplet losses with our proposed objective function.** We present the classification accuracy (Weighted F1 score) on standard datasets using XGBoost for five random train-test splits (we randomly split the data into train and test (80-20) five time, and calculated the mean and standard deviation of the accuracies). We note that the improved performance of the NPLB-trained model was consistent across different classifiers (as shown for a different dataset in §5). Results for *LiftedStruct* and *InfoNCE* are presented in Table A2.

|  | *Trad. Triplet Loss* | *N-Pair* | *MDR* | *Distance Swap* | *NPLB Objective (Ours)* |
|---|---|---|---|---|---|
| *MNIST* | $0.9859 \pm 0.0009$ | $0.9843 \pm 0.0003$ | $0.9886 \pm 0.0009$ | $0.9891 \pm 0.0003$ | $\mathbf{0.9954 \pm 0.0003}$ |
| *Fashion MNIST* | $0.9394 \pm 0.001$ | $0.9586 \pm 0.003$ | $0.9557 \pm 0.001$ | $0.9536 \pm 0.001$ | $\mathbf{0.9664 \pm 0.001}$ |
| *CIFAR10* | $0.8036 \pm 0.028$ | $0.7963 \pm 0.034$ | $0.8152 \pm 0.027$ | $0.8285 \pm 0.022$ | $\mathbf{0.8475 \pm 0.025}$ |

## 5    IMPROVING PATIENT REPRESENTATION LEARNING

In this section, we aim to demonstrate the potential and implications of our approach on a more complex dataset in three steps: First, we show that deep metric learning improves upon current

state-of-the-art patient embedding models (§5.1). Next, we provide a comparison between NPLB, Distance Swap and the traditional triplet loss formulations (§5.2). Lastly, we apply our methodology to predict health risks of currently healthy subjects from a single time point (§5.3). We focus on presenting results for the female subjects due to space limitations. We note that results on male subjects are very similar to the female population, as presented in Appendix I.

## 5.1 DEEP METRIC LEARNING FOR BETTER PATIENT EMBEDDINGS

Healthcare datasets are considerably different than those in other domains. Given the restrictions on sharing health-related data (as stipulated by laws such as those defined under the Health Insurance Portability and Accountability Act - HIPAA), most DL-based models are developed and tested on proprietary in-house datasets, making comparisons and benchmarking a major hurdle (Evans, 2016). This is in contrast to other areas of ML which have established standard datasets (e.g. ImageNet or GLUE (Wang et al., 2018a)). To show the feasibility of our approach, we present the effectiveness of our methodology on the United Kingdom Biobank (UKB) (Bycroft et al., 2018): a large-scale ($\sim 500K$ subjects) complex *public* dataset, showing the potential of UKB as an additional benchmarking that can be used for developing and testing future DL models in the healthcare domain. UKB contains deep genetic and phenotypic data from approximately 500,000 individuals aged between 39-69 across the United Kingdom, collected over many years. We considered patients' lab tests and their approximated activity levels (e.g. moderate or vigorous activity per week) as predictors (complete list of features used is shown in Appendix M), and their doctor-confirmed conditions and medication history for determining labels. Specifically, we labeled a patient as "unhealthy" if they have confirmed conditions or take medication for treating a condition, and otherwise labeled them as "apparently-healthy". We provide a step-by-step description of our data processing in Appendix F.

A close analysis of the UKB data revealed large in-class variability of test ranges, even among those with no current or prior confirmed conditions (the "apparently-healthy" subjects). Moreover, the overall distribution of key metrics are very similar between the unhealthy and apparently-healthy patients (visualized in Appendix G). As a result, we hypothesized that there exists a *continuum* among patients' health states, leading to our idea that a similarity-based *learned* embedding can represent subjects better than other representations for downstream tasks. This idea, in tandem with our assumption of intricate nonlinear relationships among features, naturally motivated our approach of *deep metric learning*: our goal is to train a model that learns a metric for separating patients in space, based on their similarities and current confirmed conditions (labels). Due to our assumptions, we used the apparently-healthy patients initially as *anchor* points between the two ends of the continuum (the very unhealthy and healthy). However, this formulation necessitates identifying a more "reliable" healthy group, often referred to as the *bona fide* healthy (BFH) group (Cohen et al., 2021)[2].

To find the BFH population, we considered all patients whose *key* lab tests for common conditions fall within the clinically-normal values. These markers are: *Total Cholesterol, HDL Cholesterol, LDL Cholesterol, Triglycerides, Fasting Glucose, HbA1c, C-Reactive Protein.*; we refer to this set of metrics as the $P_0$ metrics and provide the traditional "normal" clinical ranges in Appendix H. It is important to note that the count of the BFH population is much smaller than the apparently-healthy group ($\sim 6\%$ and $\sim 5\%$ of female and male populations, respectively). To address this issue and to keep DML as the main focus, we implemented a simple yet intuitive rejection-based sampling to generate synthetic BFH patients, though more sophisticated methods could be employed in future work. Similar to any other rejection-based sampling and given that lab results often follow a Gaussian distribution (Whyte & Kelly, 2018), we assumed that each feature follows a distribution $\mathcal{N}(\mu_x, \sigma_x)$ where $\mu_x$ and $\sigma_x$ denote the empirical mean and standard deviation of feature $x_i$ for all patients. Since BFH patients are selected if their $P_0$ biosignals fall within the clinically-normal lab ranges, we used the bounds of the clinically normal range as the accept/reject criteria. Our simple rejection-based sampling scheme is presented in Appendix L.

**Training Procedure and Model Architecture**: Before training, we split the data $70\% : 30\%$ for training and testing. In the training partition, we augment the *bona fide* population 3 folds and then generate 100K triplets of the form $(a_i, p_i, n_i)$ randomly in an offline manner. We chose to generate

---

[2]Although it is possible to further divide each group (e.g. based on conditions), we chose to keep the patients in three very general groups to show the feasibility of our approach in various health-related domains.

only 100K triplets in order to reduce training time and demonstrate the capabilities of our approach for smaller datasets. Note that in an unsupervised setting, these triplets need to be generated online via negative sample mining, but this is out of scope for this work given that we have labels *a priori* . Our model consists of the three hidden layers with two probabilistic dropout layers in between and Parametric Rectified Linear Units (PReLU) (He et al., 2015) nonlinear activations. We present a visual representation of our architecture and the dimensions in Fig. A5. We optimize the weights for minimizing our proposed NPLB objective, Eq.(3), using Adam (Kingma & Ba, 2014) for 1000 epochs with $lr = 0.001$, and employ an exponential learning rate decay ($\gamma = 0.95$) to decrease the learning rate after every 50 epochs, and set the triplet margin to $\epsilon_\circ = 1$. For simplicity, we will refer to our model as **SPHR** (Similarity-based Patient Risk modeling). In order to test the true capability of our model, all evaluations are performed on the *non-augmented* data.

**Results**: Similar to §4, we evaluated our deep-learned embeddings on its improvements for binary (unhealthy or apparently-healthy) and multi-class (unhealthy, apparently-healthy or *bona fide* healthy) classification tasks. The idea here is that if SPHR has learned to separate patients based on their conditions and similarities, then training classifiers on SPHR-produced embeddings should show improvements compared to raw data. We trained five classifiers ($k$-nearest neighbors (KNN), Linear Discriminate Analysis (LDA), a neural network (NN) for EHR (Chen et al. (2020a)), and XGBoost Chen & Guestrin (2016)) on raw data (not-transformed) and other common transformations. These transformations include a linear transformation (Principal Component analysis [PCA]), a non-linear transformation (Diffusion Maps (Coifman & Lafon (2006) [DiffMap]) and the current state-of-the-art nonlinear transformation, *DeepPatient*. DeepPatient, PCA, DiffMap, and SPHR are $\mathbb{R}^n \to \mathbb{R}^d$, where $n, d \in \mathbb{N}$ denote the initial number of features and a (reduced) mapping space, respectively, with $d = 32$ (in order to have the same dimensionality as SPHR), though various choices of $d$ yielded similar results. We present these results in Table 2 comparing classification weighted F1 score for models trained on raw EHR, linear and nonlinear transformations. We also evaluate the separability qualitatively using UMAP, as shown in Fig. A1. In all tested cases, our model significantly outperforms all other transformations, demonstrating the effectiveness of DML in better representing patients from EHR.

Table 2: **Comparison of binary and multi-label classification performance (weighted F1 score) with various representations on the *female* subjects.** We kept the same random seeds across different classifiers and randomly split the data into train and test (80-20) five time, and calculated the mean and standard deviation of the accuracies. For binary classification, we considered BFH patients as *healthy* patients (hence having binary labels). Our results show that SPHR significantly improves the classification of all tested classifiers, demonstrating better separability in space compared to raw data and state-of-the-art methods (DeepPatient).

| Model | Not-Transformed | PCA | DiffMap | DeepPatient | SPHR (Ours) |
|---|---|---|---|---|---|
| *Binary Classification* | | | | | |
| KNNs | $0.6093 \pm 0.002$ | $0.6067 \pm 0.002$ | $0.6281 \pm 0.003$ | $0.6397 \pm 0.003$ | $\mathbf{0.7180 \pm 0.002}$ |
| LDA | $0.6189 \pm 0.001$ | $0.6144 \pm 0.002$ | $0.6309 \pm 0.002$ | $0.6404 \pm 0.005$ | $\mathbf{0.7189 \pm 0.002}$ |
| NN for EHR | $0.6214 \pm 0.006$ | $0.6127 \pm 0.006$ | $0.6228 \pm 0.003$ | $0.6315 \pm 0.001$ | $\mathbf{0.7142 \pm 0.003}$ |
| XGBoost | $0.6113 \pm 0.003$ | $0.6088 \pm 0.002$ | $0.6196 \pm 0.005$ | $0.6311 \pm 0.003$ | $\mathbf{0.7223 \pm 0.002}$ |
| *Multi-Label Classification* | | | | | |
| KNNs | $0.5346 \pm 0.002$ | $0.5341 \pm 0.002$ | $0.5646 \pm 0.006$ | $0.5740 \pm 0.004$ | $\mathbf{0.6534 \pm 0.002}$ |
| LDA | $0.5501 \pm 0.002$ | $0.5432 \pm 0.001$ | $0.5510 \pm 0.007$ | $0.5637 \pm 0.003$ | $\mathbf{0.6640 \pm 0.001}$ |
| NN for EHR | $0.5374 \pm 0.004$ | $0.5473 \pm 0.003$ | $0.5541 \pm 0.004$ | $0.5774 \pm 0.004$ | $\mathbf{0.6667 \pm 0.002}$ |
| XGBoost | $0.5190 \pm 0.003$ | $0.5082 \pm 0.003$ | $0.5390 \pm 0.002$ | $0.5688 \pm 0.002$ | $\mathbf{0.6642 \pm 0.002}$ |

Table 3: **Comparison of SOTA Triplet losses with our proposed objective on UK Biobank data (male and female patients).** We quantitatively evaluate embeddings by measuring classification accuracy (weighted F1 scores) of XGBoost on five random train-test splits of the UK Biobank for each gender. As shown below, our model significantly outperforms traditional and distance swap loss in both binary and multi-class classification, demonstrating its potential in real-world applications, such as healthcare applications. Results for *LiftedStruct* and *InfoNCE* are presented in Table A2.

| | Trad. Triplet Loss | N-Pair | MDR | Distance Swap | NPLB Objective (Ours) |
|---|---|---|---|---|---|
| *Females*: Binary | $0.6592 \pm 0.002$ | $0.6727 \pm 0.004$ | $0.6612 \pm 0.003$ | $0.6041 \pm 0.002$ | $\mathbf{0.7223 \pm 0.002}$ |
| *Females*: Multi-Label | $0.5874 \pm 0.001$ | $0.6064 \pm 0.002$ | $0.6047 \pm 0.005$ | $0.5416 \pm 0.004$ | $\mathbf{0.6642 \pm 0.002}$ |
| *Males*: Binary | $0.7174 \pm 0.001$ | $0.7423 \pm 0.005$ | $0.7208 \pm 0.002$ | $0.6563 \pm 0.003$ | $\mathbf{0.8160 \pm 0.003}$ |
| *Males*: Multi-Label | $0.6861 \pm 0.003$ | $0.6961 \pm 0.002$ | $0.6964 \pm 0.002$ | $0.6628 \pm 0.002$ | $\mathbf{0.7845 \pm 0.003}$ |

## 5.2 *NPLB* SIGNIFICANTLY IMPROVES LEARNING ON COMPLEX DATA

One of our main motivations for modifying the triplet object was to improve model performance on more complex datasets with larger in-class variability. To evaluate the improvements provided by our *NPLB* objective, we perform the same analysis as in §4, but this time on the UKB data. Table 3 demonstrate the significant improvement made by our simple modification to the traditional triplet loss, further validating our approach and formulation experimentally.

## 5.3 PREDICTING HEALTH RISKS FROM A SINGLE LAB VISIT

Table 4: **The percentage of *apparently-healthy* female subjects who later develop conditions within each predicted risk group**. For the same number of *healthy subjects*, we use embeddings from each method to classify patients into three health risk groups using the provided health risk definition [Normal (N), Lower Risk (LR), and Higher Risk (HR)]. To validate each model's risk prediction, we use future health assessment to calculate the percentage of subjects in each risk group who later developed Cancer, Diabetes or Other Serious Conditions. Among all methods (top three shown), SPHR-predicted *Normal* and *Higher Risk* patients developed the fewest and most conditions, respectively, as desired.

| | P0 (Not-Transformed) | | | DeepPatient | | | SPHR (Ours) | | |
|---|---|---|---|---|---|---|---|---|---|
| | Normal | LR | HR | Normal | LR | HR | Normal | LR | HR |
| **Predicted Healthy Subjects in Each Risk Group** | | | | | | | | | |
| *Percentage* | 74.71% | 5.82% | 16.26% | 77.19% | 7.61% | 15.2% | 68.35% | 6.94% | 24.71% |
| **Validation on Future Diagnosis** | | | | | | | | | |
| *Cancer* | 3.17% | 0.92% | 2.83% | 3.21% | <0.1% | <0.1% | 1.67% | 2.85% | 3.22% |
| *Diabetes* | 1.26% | 1.85% | 1.41% | 0.51% | <0.1% | <0.1% | 0.50% | <0.1% | 4.80% |
| *Other Serious Cond* | 9.3% | 5.66% | 9.29% | 8.77% | 4.98% | 6.81% | 1.47% | 8.33% | 11.65% |

**Definition of Single-Time Health Risk**: Predicting patients' future health risks is a challenging task, especially when using only a single lab visit. As described in §2, all current models use multiple assessments for predicting health risks of a patient; however, these approaches ignore a large portion of the population who do not return for additional check-ups. Motivated by the definition of risk in other fields (e.g. risk of re-identification of anonymized data (Ito & Kikuchi, 2018)), we provide a simple and intuitive distance-based definition of health risk to address the mentioned issues and that is well suited for DML embeddings. Given the simplicity of our definition and due to space constraints, we describe the definition below and outline the mathematical framework in Appendix B.

We define the *health distance* as the euclidean distance between a subject and the reference *bona fide* healthy (BFH) subject. Many studies have shown the large discrepancy of lab metrics among different age groups and genders (Cohen et al., 2021). To account for these known differences, we identify a *reference* vector which is the median BFH subject from each age group per gender. Moreover, for simplicity and interpretability, we define *health risk* as discrete groups using the known BFH population: For each stratification $g$ (age and sex), we identify two BFH subjects who are at the 2.5 and 97.5 percentiles (giving us the inner 95% of the distribution), and calculate their distance to the corresponding reference vector, This gives us a distance interval $[t_{2.5}^g, t_{97.5}^g]$. In a group $g$, any new patient whose distance to the reference vector falls inside the corresponding $[t_{2.5}^g, t_{97.5}^g]$ is considered to be "Normal". Similarly, we identify the $[t_1^g, t_{99}^g]$ intervals (corresponding to the inner 98% of BFH group); any new patient whose health distance is within $[t_1^g, t_{99}^g]$ but not in $[t_{2.5}^g, t_{97.5}^g]$ is considered to be in the "Lower Risk" (LR) group. Lastly, any patient with health distance outside of these intervals is considered to have "Higher Risk" (HR). The UKB data is a good candidate for predicting potential health risks, given that it includes subsequent follow-ups where a subset of patients are invited for a repeat assessment. The first follow-up was done between 2012-2013 and it included approximately 20,000 individuals (Littlejohns et al., 2019) (25× reduction with many measurements missing). Based on our goal of predicting risk from a single visit, we *only* include the patients' first visits for modeling, and use the 2012-2013 follow-ups for *evaluating* the predictions.

**Results**: We utilized the single-time health risk definition to predict a patient's future potentials of health complications. To demonstrate the versatility of our approach, we predicted *general* health risks that were used for all health conditions available (namely *Cancer, Diabetes and Other Serious Conditions*); however, we hypothesize that constraining health risks based on specific conditions will improve risk predictions. We considered five methods for assigning a risk group to each

patient: (i) Euclidean distance on raw data (preprocessed but not transformed), (ii) Mahalanobis distance on pre-processed data, (iii) Euclidean distance on the key metrics ($P_0$) for the available conditions(described in section 5.1), therefore hand-crafting features and reducing dimensionality in order to achieve an upper bound performance for most traditional methods (though this will not be possible for all diseases). The last two methods we consider are deep representation learning methods: (iv) DeepPatient and (v) SPHR embeddings (proposed model). We use the Euclidean metric for calculating the distance between deep-learned representations of (iv) and (v). For all approaches, we assigned *all* patients to one of the three risk groups using biosignals from a single visit, and calculated the percentage of patients who developed a condition in the immediate next visit. Intuitively, patients who fall under the "Normal" group should have fewer confirmed cases compared to subjects in "Lower Risk" or "Higher Risk" groups. Table 4 shows the results for the top three methods, with our approach consistently matching the intuitive criteria: among the five methods, SPHR-predicted patients in the *Normal* risk group have the fewest instance of developing future conditions, while the ones predicted as *High* risk have the highest instance of developing future conditions (Table 4).

## 6 CONCLUSION AND DISCUSSION

We present a simple and intuitive variation of the traditional triplet loss (*NPLB*) which regularizes the distance between positive and negative samples based on the distance between anchor and negative pairs. To show the general applicability of our methods and as an initial validation step, we tested our model on three standard benchmarking datasets (MNIST, Fashion MNIST and CIFAR-10) and found that our NPLB-trained model produced better embeddings. To demonstrate the real world impact of DML, such as our proposed framework called SPHR, we applied our methodology on the UKB to classify patients and predict future health risk for the current healthy patients, using only a single time point. Motivated by risk prediction in other domains, we provide a distance-based definition of health risk. Utilizing this definition, we partitioned patients into three health risk groups (*Normal, Lower Risk* and *Higher Risk*). Among all methods, SPHR-predicted *Higher Risk* healthy patients had the highest percentage of actually developing conditions in the next visit, while SPHR-predicted *Normal* patients had the lowest instances, which is desired. Although the main point of our work focused on modifying the objective function, a limitation of our work is the simple triplet sampling that we employed, particularly when applied to healthcare. We anticipate additional improvement gains by employing online triplet sampling or extending our work to be self-supervised (Chen et al., 2020b; Oord et al., 2018b; Wang et al., 2021).

The implications of our work are threefold: (1) Our proposed objective has the potential of improving existing triplet-based models without requiring additional sample mining or computationally intensive operations. We anticipate that combining our work with existing triplet sampling can further improve model learning and results. (2) Models for predicting patients' health risks are nascent and often require time-series data. Our experiments demonstrated the potential improvements gained by developing DML-based models for learning patient embeddings, which in turn can improve patient care. Our results show that more general representation learning models are valuable in pre-processing EHR data and producing deep learned embeddings that can be used (or fine-tuned) for more specific downstream analyses. We believe additional analysis on the learned embedding space can prove to be useful for various tasks. For example, we show that there exists a relationship between distances in the embedded space and the time to develop a condition, which we present in Appendix C. The rapid growth of healthcare data, such as EHR, necessitates the use of large-scale algorithms that can analyze such data in a scalable manner (Evans, 2016). Currently, most applications of ML in healthcare are formulated for small-scale studies with proprietary data, or use the publicly-available MIMIC dataset (Miotto et al., 2016; Johnson et al., 2016), which is not as large-scale and complex as the UKB. Similar to our work, we believe that future DL models can benefit from using the UKB for development and benchmarking. (3) Evaluating health risk based on a single lab visit can enable clinicians to flag high-risk patients early on, potentially reducing the number (and the scope) of costly tests and significantly improving care for the most vulnerable individuals in a population.

REPRODUCIBILITY

Our code package and tutorial notebooks are all publicly available at on the authors' Github at: <revealed after the double blind reviews>, and we will actively monitor the repository for any issues or improvements suggested by the users. Moreover, we have designed our Appendix to be a comprehensive guide for reproducing our results and experiments as well. Our Appendix includes all used features from the UK Biobank (for male and female patients), a complete list of model parameters (for classification models), detailed definition of single-time health risk as well as pseudocode and description of architectures we developed for our experiments.

ACKNOWLEDGMENTS

Will be added after double blind reviews.

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

# Appendix

## APPENDIX A    RUNTIME AND SCALING ANALYSIS OF SPHR

To measure the scalability of SPHR, we generated a random dataset consisting of 100,000 samples with 64 linearly-independent (also known as "informative") features and 10 classes, resulting in a matrix $X \in \mathbb{R}^{100000 \times 64}$. From this dataset, we then randomly generated varying number of triplets following the strategy described in the main manuscript and in Hoffer & Ailon (2015), representing a computationally intensive case (as opposed to more intricate and faster mining schemes). The average of five mining runs is presented as "Avg. Mining Time" column in Table A1. Next, we trained SPHR on the varying number of triplets five times using (1) A Google Compute Engines with 48 logical cores (CPU) (2) A Google Virtual Machine equipped with one NVIDIA V100 GPU (referred to as *GPU*). The average training times for CPUs and GPUs are shown in Table A1.

Table A1: **Average training time of the proposed deep learning model (SPHR)**. All experiments are done under the same computational The times shown below are the average of 5 runs with all identical settings.

| Number of Triplets | Avg. CPU Training Time | Avg. GPU Training Time | Avg. Mining Time |
|---|---|---|---|
| 1,000 | 8.28 Mins | 3.95 Mins | <0.01 Mins |
| 5,000 | 15.07 Mins | 5.96 Mins | 0.01 Mins |
| 10,000 | 31.02 Mins | 7.60 Mins | 0.12 Mins |
| 50,000 | 76.24 Mins | 16.79 Mins | 0.65 Mins |
| 100,000 | 133.26 Mins | 27.15 Mins | 1.27 Mins |
| 500,000 | 512.75 Mins | 94.79 Mins | 5.58 Mins |

## APPENDIX B    MATHEMATICAL DEFINITION OF SINGLE-TIME HEALTH RISK

In this section, we aim to provide a mathematical definition of health risk that can measure similarity between a new cohort and an existing *bona fide* healthy population without requiring temporal data. This definition is inspired by the formulation of risk in other domains, such as defining the risk of re-identification of anonymized data (Ito & Kikuchi, 2018). We first define the notion of a "health distance", and use that to formulate thresholds intervals which enable us to define health risk as a mapping between continuous values and discrete risk groups.

**Definition Appendix B.1.**  Given a space $X$, a *bona fide* healthy population distribution $B$ and a new patient $p$ (all in $X$), the **health score** $s_n$ of $p$ is:

$$s_q(p) = d(P_q(B), p)$$

where $P_q(H)$ denotes the $q^{\text{th}}$ percentile of $B$, and $d(\cdot)$ refers to a metric defined on $X$ (potentially a pseudo metric).

Definition Appendix B.1 provides a measure of distance between a new patient and an existing reference population, which can be used to define similarity. For example, if $d(\cdot)$ is the Euclidean distance, then the similarity between patient $x$ and $y$ is $sim(x, y) = \frac{1}{1+d(x,y)}$. Next, using this notion of distance (and similarity), we define risk thresholds that allow for grouping of patients.

**Definition Appendix B.2.**  A threshold interval $I_q = [t_q^l, t_q^u]$ is defined as the distance between the vectors providing the inner $q$ percent of a distribution $H$ and the median value. Let $n = (100 - q)/2$, then we have $I_q$ as:

$$I_q = [t_q^l, t_q^u] = [d(P_n(H), P_{50}(H)), d(P_{q+n}(H), P_{50}(H))].$$

Additionally, for the sake of interpretability and convenience, we can define **health risk groups** using *known $I_q$* values which we define below.

**Definition Appendix B.3.**  Let $M : \mathbb{R}^d \to V \cup \{\eta\}$ where $d$ denotes the number of features defining a patient, with $\eta$ being discrete group and $V$ denoting pre-defined risk groups based on $k \in \mathbb{N}$ many

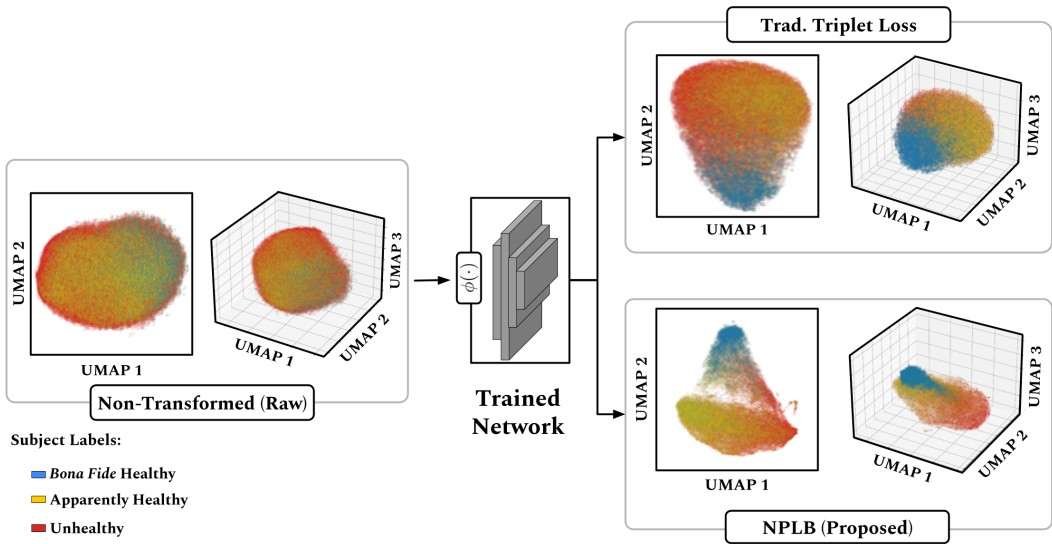

Figure A1: **Qualitative assessment of our approach on UKB female patients.** The NPLB-trained network better separates subjects, resulting in significant improvements in classification of patients, as shown in Table 3. Note the continuum of patients for both metric learning techniques, especially among the apparently-healthy patients (in yellow).

intervals (i.e. $|V| = |I_q| = k$). Using the same notion of health distance $s_n$ and threshold interval $I_q$ as before, we define a patient $p$'s health risk group as:

$$M(p) = \begin{cases} V_q & \text{if } s_q(p) \in I_q \\ \eta & \text{otherwise} \end{cases}.$$

We utilize our deep metric learning model and the definitions above in tandem to predict health risks. That is, we first produce embeddings for all patients using our learned nonlinear operator $G$, and then use the distance between the *bona fide* healthy (BFH) population and new patients to assign them a risk group. Mathematically, given the set of BFH population for all groups, i.e.

$$B_{All} = \{B_{[36,45]}, B_{[46,50]}, B_{[51,55]}, B_{[56,60]}, B_{[61,65]}, B_{[66,75]}\},$$

we take the reference value *per age group* to be

$$\tilde{r}_{age} = G(P_{50}(B_{age})),$$

where $B_{age}$ denotes the BFH population for the age group $age$. Then, using definition Appendix B.2, we define:

$$N_{age} = I_{95}^{age} = [d(P_{2.5}(B_{age}), P_{50}(B_{age})), d(P_{97.5}(B_{age}), P_{50}(B_{age}))] \tag{A.4a}$$

$$LR_{age} = I_{98}^{age} = [d(P_1(B_{age}), P_{50}(B_{age})), d(P_{99}(B_{age}), P_{50}(B_{age}))]. \tag{A.4b}$$

Note that $N_{age} \subset LR_{age}$. Lastly, using the intervals defined in Eq. (A.4), we define the mapping $M$ as shown in Eq. (A.5).

$$M_{age}(p) = \begin{cases} \text{Normal} & \text{if } s_n(G(p)) = d(G(p), \tilde{r}_{age}) \in N_{age} \\ \text{Lower Risk} & \text{if } s_n(G(p)) = d(G(p), \tilde{r}_{age}) \in LR_{age} \backslash N_{age} \\ \text{Higher Risk} & \text{otherwise} \end{cases}. \tag{A.5}$$

## APPENDIX C    PREDICTING PATIENT'S HEALTH RISK IN TIME

Given the performance of our DML model in classifying subjects and health risk assessment, we hypothesized that we can retrieve a relationship between spatial distance (in the embedded space)

and a patient's time to develop a condition using only a single lab visit. Let us assume that subjects start from a "healthy" point and move along a trajectory (among many trajectories) to ultimately become "unhealthy" (similar to the *principle of entropy*). In this setting, we hypothesized that our model maps patients in space based on their potential of moving along the trajectory of becoming unhealthy. To test our hypothesis, we designed an experiment to use our distance-based definition of health risk, and the NPLB embeddings to further stratify patients based on the *immediacy of their health risk* (time). More specifically, we investigated the correlation between spatial locations of the *currently-healthy* patients in the embedded space and the time to which they develop a condition.

Similar to the health risk prediction experiment in the main manuscript, we computed the distance between each patient's embedding and the corresponding reference value in the ultra healthy population (refer to the main manuscript for details on this procedure). We then extracted all healthy patients at the time of first visit who returned in (2012-2013) and 2014 (and after) for reassessment or imaging visits. It is important to again note that there is a significant drop in the number of returning patients for subsequent visits. Among the retrieved patients, we calculated the number of individuals who developed *Cancer, Diabetes* or *Other Serious Conditions*. We found strong negative correlations between the calculated health score (distances) and time of diagnosis for Other Serious Conditions ($r = -0.72, p = 0.00042$) and Diabetes ($r = -0.64, p = 0.0071$), with Cancer having the least correlation $r = -0.20$ and $p = 0.025$. Note that the lab tests used as predictors are associated with diagnosing metabolic health conditions and less associated with diagnosing cancer, which could explain the low correlation between health score and time of developing cancer. These results indicate that the metric learned by our model accounts for the immediacy of health risk, mapping patient's who are at a higher risk of developing health conditions farther from those who are at a lower risk (hence the negative correlation).

## APPENDIX D : NPLB CONDITION IN MORE DETAIL

In this section, we aim to take a closer look at the minimizer of our proposed objective. Using the same notations as in the main manuscript, we define the following variables for convenience:

$$\delta_+ \triangleq d(\phi_a, \phi_p)$$
$$\delta_- \triangleq d(\phi_a, \phi_n)$$
$$\rho \triangleq d(\phi_p, \phi_n)$$

With this notation, we can rewrite our proposed *No Pairs Left Behind* objective as:

$$\mathcal{L}_{NPLB} = \frac{1}{N} \sum_{(p_i, a_i, n_i) \in T}^{N} [\delta_+ - \delta_- + \epsilon_0]^+ + (\rho - \delta_-)^2. \tag{A.7}$$

Note that the since $[\delta_+ - \delta_- + \epsilon_0]^+ \geq 0, \mathcal{L}_{NPLB} = 0$ *if and only* the summation of each term is identically zero. This yields the following relation:

$$-(\rho - \delta_-)^2 = [\delta_+ - \delta_- + \epsilon_0]^+ \tag{A.8}$$

which, considering the real solutions, is only valid if $\rho = \delta_-$, and if $\delta_- > \delta_+ + \epsilon_0$, and therefore $\rho > \delta_+ + \epsilon_0$. As a result, the regularization term enforces that the distance between the positive and the negative to be at least as much $\delta_+ + \epsilon_0$, leading to denser clusters that are better separated from other classes in space.

NPLB can be very easily implemented using existing implementations in standard libraries. As an example, we provide a Pytorch-like pseudo code showing the implementation of our approach:

```python
from typing import Callable, Optional
import torch

class NPLBLoss(torch.nn.Module):

```

```python
6    def __init__(self,
7                 triplet_criterion: torch.nn.TripletMarginLoss,
8                 metric: Optional[Callable[
9                     [torch.Tensor, torch.Tensor],
10                    torch.Tensor]] = torch.nn.functional.pairwise_distance
     ):
11       """Initializes the instance with backbone triplet and distance metric
         ."""
12       super().__init__()
13       self.triplet = triplet_criterion
14       self.metric = metric
15
16   def forward(self, anchor: torch.Tensor, positive: torch.Tensor,
17               negative: torch.Tensor) -> torch.Tensor:
18       """Forward method of NPLB loss."""
19
20       # Traditional triplet as the first component of the loss function.
21       triplet_loss = self.triplet(anchor, positive, negative)
22       # This
23       positive_to_negative = self.metric(positive, negative, keepdim=True)
24       anchor_to_negative = self.metric(anchor, negative, keepdim=True)
25       # Here we use the reduction to be 'mean', but it can be any kind
26       # that the DL library would support.
27       return triplet_loss + torch.mean(
28           torch.pow((positive_to_negative - anchor_to_negative), 2))
```

Listing 1: Pytorch implementation of NPLB.

## APPENDIX E   LIFTEDSTRUCT, N-PAIRS LOSS, AND INFONCE

In this section, we provide a brief description of three popular deep metric learning models that are related to our work. We also describe the implementations used for these models, and present a complete comparison of all methods on all datasets in this work. In *LiftedStruct* (Song et al., 2015), the authors propose to take advantage of the full batch for comparing pairs as opposed to traditional approaches where positive and negative pairs are pre-defined for an anchor. The authors describe their approach as "lifting" the vector of pairwise distances for each batch to the matrix of pairwise distances. *N-Pair Loss* (Sohn, 2016) is a generalization of the traditional triplet loss which aims to address the "slow" convergence of traditional triplet models through considering $N - 1$ negative examples instead of the one negative pair considered in the traditional approach. *InfoNCE* (Oord et al., 2018a) is a generalization of N-pair loss that is also known as the normalized temperature-scaled cross entropy loss (NT-Xent). This loss aims to maximize the agreement between positive samples. Both N-Pair and InfoNCE loss relate to our work due to their formulation of the metric learning objective that are closely related to Triplet loss. To compare our approach against these algorithms, we leveraged the widely used Pytorch Metric Learning (PML) package (Musgrave et al., 2020). The complete results of our comparisons on all tested datasets are shown below in Table A2.

Table A2: **Comparison of state-of-the-art (SOTA) triplet losses with our proposed objective function (complete version of Tables 1 and 3 of the main manuscript).** The classifications were done on the embeddings using XGBoost for five different train-test splits, with the average Weighted F1 score reported below. We note that the improved performance of the NPLB-trained model was consistent across different classifiers. The UKB results are for the multi-class classification.

|  | *MNIST* | *FashionMNIST* | *CIFAR10* | *UKB (Females)* | *UKB (Males)* |
|---|---|---|---|---|---|
| *Trad. Triplet Loss* | $0.9859 \pm 0.0009$ | $0.9394 \pm 0.001$ | $0.8036 \pm 0.028$ | $0.5874 \pm 0.001$ | $0.6861 \pm 0.003$ |
| *N-Pair* | $0.9863 \pm 0.0003$ | $0.9586 \pm 0.003$ | $0.7936 \pm 0.034$ | $0.6064 \pm 0.004$ | $0.6961 \pm 0.002$ |
| *LiftedStruct* | $0.9853 \pm 0.0007$ | $0.9495 \pm 0.002$ | $0.7946 \pm 0.041$ | $0.5994 \pm 0.003$ | $0.6989 \pm 0.004$ |
| *MDR* | $0.9886 \pm 0.0003$ | $0.9557 \pm 0.003$ | $0.8152 \pm 0.027$ | $0.6047 \pm 0.005$ | $0.6964 \pm 0.002$ |
| *InfoNCE* | $0.9858 \pm 0.0002$ | $0.9581 \pm 0.004$ | $0.8039 \pm 0.026$ | $0.6103 \pm 0.002$ | $0.6816 \pm 0.003$ |
| *Distance Swap* | $0.9891 \pm 0.0003$ | $0.9536 \pm 0.001$ | $0.8285 \pm 0.022$ | $0.5416 \pm 0.004$ | $0.6628 \pm 0.002$ |
| *NPLB (Ours)* | $\mathbf{0.9954 \pm 0.0003}$ | $\mathbf{0.9664 \pm 0.001}$ | $\mathbf{0.8475 \pm 0.025}$ | $\mathbf{0.6642 \pm 0.002}$ | $\mathbf{0.7845 \pm 0.003}$ |

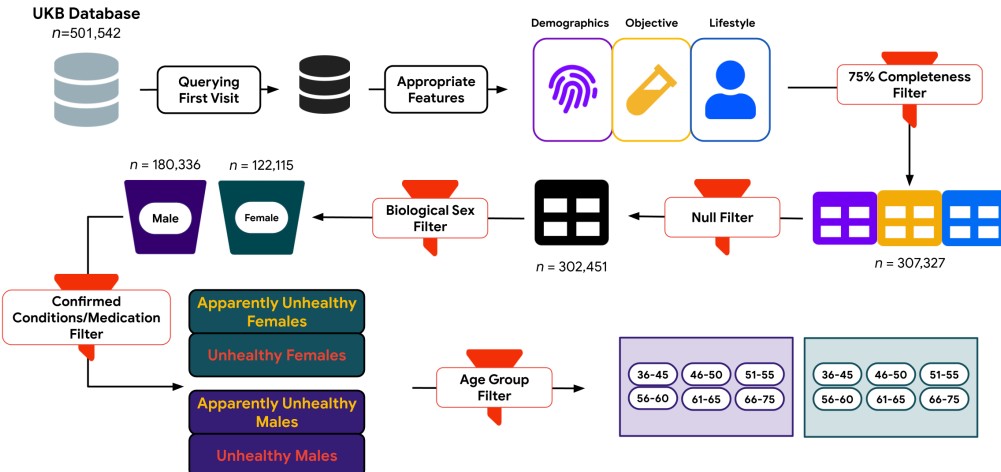

Figure A2: **Visualization of the data processing scheme described in Section Appendix F**.

## APPENDIX F    UK BIOBANK DATA PROCESSING

Given the richness and complexity of the UKB and the scope of this work, we subset the data to include patients' age and gender (demographics), numerous lab metrics (objective features), Metabolic Equivalent Task (MET) scores for vigorous/moderate activity and self-reported hours of sleep (lifestyle) (complete list of features in Appendix M). Additionally, we leverage doctor-confirmed conditions as well as current medication to assess subjects' health (assigning labels and not used as predictors).

After selecting these features, we use the following scheme to partition the subjects (illustrated in Fig. A2):

1. Ensure all features are at least 75% complete (i.e. at least 75% of patients have a non-null value for that feature)

2. Exclude subjects with any null values

3. Split resulting data according to biological sex (male or female), and perform quantile normalization (as in Cohen et al. (2021))

4. For each sex, partition patients into "unhealthy" (those who have at least one doctor-confirmed health condition or take medication for treating a serious condition) and "apparently healthy" population (who do not have any serious health conditions and do not take medications for treating such illnesses). This data is used for training our neural network.

5. Split patients into six different age groups: Each age group is constructed so that the number of patients in each group is on the same order, while the bias in the data is preserved (age groups are shown in Fig. A2). These age groups are used to determine age-specific references at the time of risk prediction.

## APPENDIX G    : SIMILARITY OF DISTRIBUTIONS FOR KEY METRICS AMONG PATIENTS

Although lab metric ranges seem very different at a first glance, look at the age-stratified ranges of tests show similarity among the apparently healthy and unhealthy patients. Additionally, if we further stratify the data based on lifestyle, the similarities between the two health groups becomes even more evident. The additional filtering is as follows: We identify the median sleep hours per group as well as identifying "active" and "less active" individuals. We define active as someone who is moderately active for 150 minutes or vigorously active for 75 minutes per week. We use the additional filtering to further stratify patients in each age group. Below in Fig. A3 and A4 we show examples of these

similarities for two age groups chosen at random. These results motivated our approach in identifying the *bona fide* healthy population to be used as reference points.

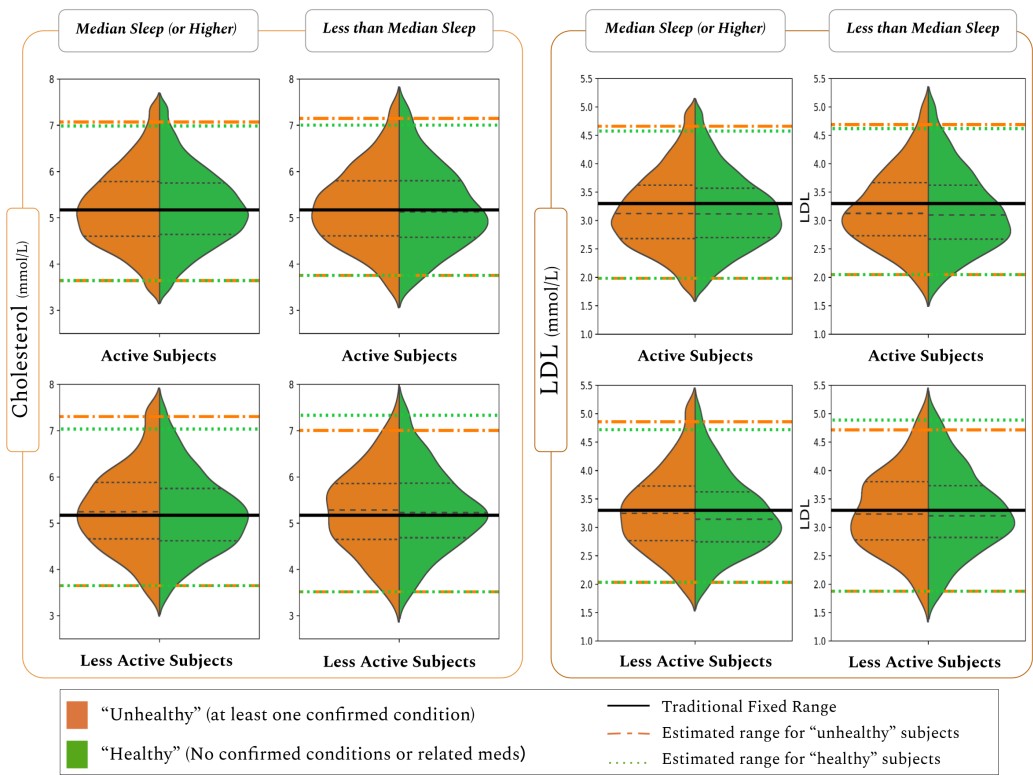

Figure A3: **Distribution similarity of key lab metrics between apparently-healthy and unhealthy female patients.** We present the violin plot for Total Cholestrol (left) and LDL Cholestrol (right) for patients between the ages of 36-45 (chosen at random). This figure aims to illustrate the similarity between these distributions based on lifestyle and age. That is, by stratifying the patients based on their sleep and activity we can see that health status alone can not separate the patients well, given the similarity in the signals.

## APPENDIX H    NORMAL RANGES FOR KEY METRICS

Below we provide a list of the current "normal" lab ranges for key metrics that determined the *bona fide* healthy population:

| Key Biomarker | Gender Specific? | Range for Males | Range for Females | Reference |
|---|---|---|---|---|
| **Total Cholesterol** | No | $\leq 5.18 \frac{mmol}{L}$ | $\leq 5.18 \frac{mmol}{L}$ | Link to Reference 1, Link to Reference 2 |
| **HDL** | Yes | $\geq 1 \frac{mmol}{L}$ | $\geq 1.3 \frac{mmol}{L}$ | Link to Reference 1, Link to Reference 2 |
| **LDL** | No | $\leq 3.3 \frac{mmol}{L}$ | $\leq 3.3 \frac{mmol}{L}$ | Link to Reference 1, Link to Reference 2 |
| **Triglycerides** | No | $\leq 1.7 \frac{mmol}{L}$ | $\leq 1.7 \frac{mmol}{L}$ | Link to Reference 1, Link to Reference 2 |
| **Fasting Glucose** | No | $\in [70, 100] \frac{mg}{dL}$ | $\in [70, 100] \frac{mg}{dL}$ | Link to Reference 3 |
| **HbA1c** | No | $< 42 \frac{mmol}{mol}$ | $< 42 \frac{mmol}{mol}$ | Link to Reference 4 |
| **C-Reactive Protein** | No | $< 10 \frac{mg}{L}$ | $< 10 \frac{mg}{L}$ | Link to Reference 5 |

## APPENDIX I    : PERFORMANCE OF SPHR ON MALE SUBJECTS

In this section, we present the results of the experiments in the main manuscript (which were done for female patients) for the male patients. The classification results are presented in Tables A3, A4, A5, and A6, and the health risk predictions are shown in Table A7.

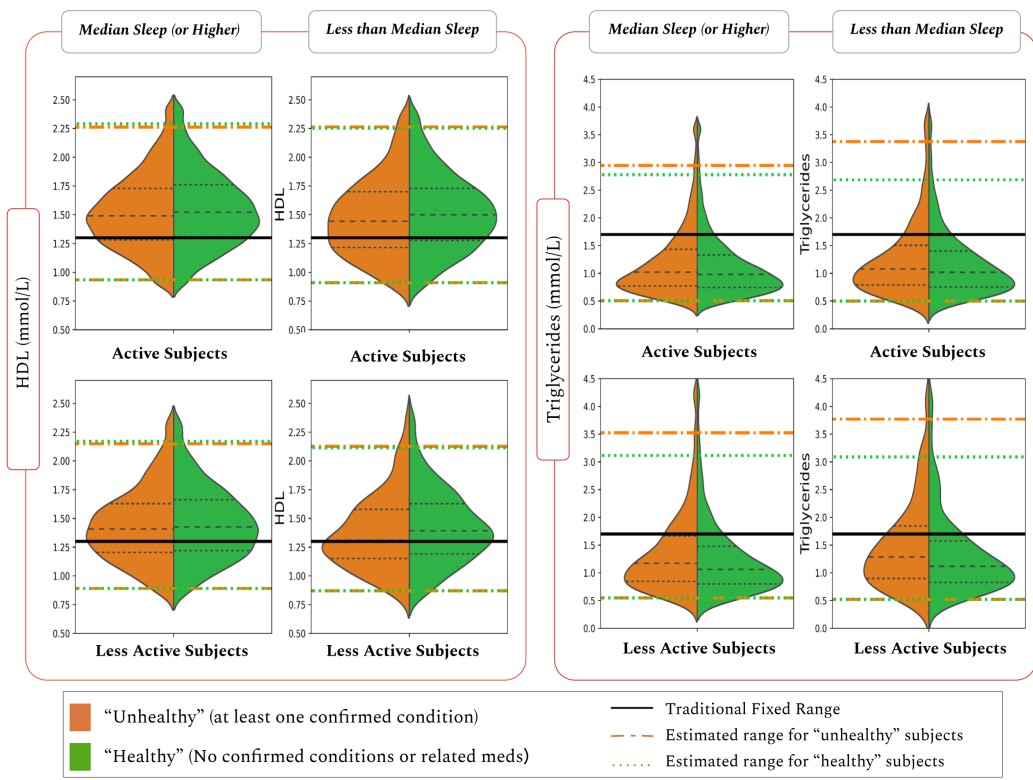

**Figure A4: Distribution similarity of key lab metrics between apparently-healthy and unhealthy female patients.** We present the violin plot for HDL Cholestrol (left) and Triglycerides (right) for patients between the ages of 55-60 (age group chosen at random). This figure aims to illustrate the similarity between these distributions based on lifestyle and age. That is, by stratifying the patients based on their sleep and activity we can see that health status alone can not separate the patients well, given the similarity in the signals.

**Table A3: Comparison of *binary* classification performance (*weighted* F1 score) with various representations on the male patients.** In this case, we consider the *bona fide* healthy patients as healthy patients and train each model to predict binary labels. We keep the same random seeds across different classifiers, and for the supervised methods, we randomly split the data into train and test (80-20) five time, and calculate the mean and standard deviation of the accuracies. Our model significantly improves the classification all tested classifiers, demonstrating better separability in space compared to raw data and state-of-the-art method (DeepPatient).

| Model | Not-Transformed | ICA | PCA | DeepPatient | SPHR (Ours) |
|---|---|---|---|---|---|
| *KNNs* | $0.6200 \pm 0.004$ | $0.6167 \pm 0.003$ | $0.6077 \pm 0.003$ | $0.6224 \pm 0.001$ | $\mathbf{0.8163 \pm 0.002}$ |
| *LDA* | $0.6275 \pm 0.005$ | $0.6227 \pm 0.004$ | $0.6227 \pm 0.003$ | $0.6385 \pm 0.002$ | $\mathbf{0.8141 \pm 0.002}$ |
| *NN for EHR* | $0.5926 \pm 0.014$ | $0.6301 \pm 0.018$ | $0.6105 \pm 0.021$ | $0.6148 \pm 0.032$ | $\mathbf{0.8092 \pm 0.011}$ |
| *XGBoost* | $0.5975 \pm 0.004$ | $0.5804 \pm 0.004$ | $0.6157 \pm 0.004$ | $0.6101 \pm 0.004$ | $\mathbf{0.8160 \pm 0.003}$ |

## APPENDIX J  EFFECTS OF AUGMENTATION ON DML

In order to evaluate the effect of augmenting the *bona fide* population and to determine the appropriate increase fold, we trained SPHR with different levels of augmentation, and evaluated the effect of each increase fold through multi-label classification performance. More specifically, created new augmented datasets with no augmentation, $1\times$, $3\times$, $5\times$ and $10\times$ and generated the same number of triplets (as described previously) and trained SPHR. We evaluated the multi-label classification using the same approach and classifiers as before (described in the main manuscript) and present the results of XGBoost classification in Table A8. Based on our findings and considerations for computational efficiency, we chose 3x augmentation as the appropriate fold increase.

Table A4: **Comparison of *binary* classification performance (*micro* F1 score) with various representations on the male patients.** In this case, we consider the *bona fide* healthy patients as healthy patients and train each model to predict binary labels. We keep the same random seeds across different classifiers, and for the supervised methods, we randomly split the data into train and test (80-20) five time, and calculate the mean and standard deviation of the accuracies. Our model significantly improves the classification all tested classifiers, demonstrating better separability in space compared to raw data and state-of-the-art method (DeepPatient).

| Model | Not-Transformed | ICA | PCA | DeepPatient | SPHR (Ours) |
|---|---|---|---|---|---|
| KNNs | $0.6490 \pm 0.004$ | $0.6480 \pm 0.004$ | $0.6469 \pm 0.003$ | $0.6639 \pm 0.001$ | **0.8185 ± 0.002** |
| LDA | $0.6529 \pm 0.004$ | $0.6509 \pm 0.002$ | $0.6509 \pm 0.002$ | $0.6664 \pm 0.003$ | **0.8138 ± 0.002** |
| NN for EHR | $0.6345 \pm 0.003$ | $0.6419 \pm 0.003$ | $0.6380 \pm 0.003$ | $0.6527 \pm 0.005$ | **0.8176 ± 0.004** |
| XGBoost | $0.6573 \pm 0.002$ | $0.6488 \pm 0.003$ | $0.6469 \pm 0.003$ | $0.6701 \pm 0.003$ | **0.8180 ± 0.003** |

Table A5: **Comparison of *multi-label* classification accuracy (*weighted* F1 score) with various representations on the male patients.** We keep the same random seeds across different classifiers, and for the supervised methods, we randomly split the data into train and test (80-20) five time, and calculate the mean and standard deviation of the accuracies. Our model significantly improves the classification for all tested classifiers, demonstrating better separability in space compared to raw data and state-of-the-art method (DeepPatient).

| Model | Not-Transformed | PCA | ICA | DeepPatient | SPHR (Ours) |
|---|---|---|---|---|---|
| KNNs | $0.5852 \pm 0.005$ | $0.5820 \pm 0.005$ | $0.5734 \pm 0.003$ | $0.5834 \pm 0.002$ | **0.7819 ± 0.001** |
| LDA | $0.6011 \pm 0.004$ | $0.5953 \pm 0.003$ | $0.5952 \pm 0.002$ | $0.6080 \pm 0.003$ | **0.7865 ± 0.002** |
| NN for EHR | $0.5926 \pm 0.004$ | $0.5925 \pm 0.004$ | $0.5838 \pm 0.003$ | $0.5918 \pm 0.001$ | **0.7884 ± 0.005** |
| XGBoost | $0.5439 \pm 0.005$ | $0.5583 \pm 0.004$ | $0.5587 \pm 0.005$ | $0.5896 \pm 0.003$ | **0.7845 ± 0.003** |

# APPENDIX K   ADDITIONAL DETAILS ON MODEL ARCHITECTURES

## K.1   SPHR'S NEURAL NETWORK

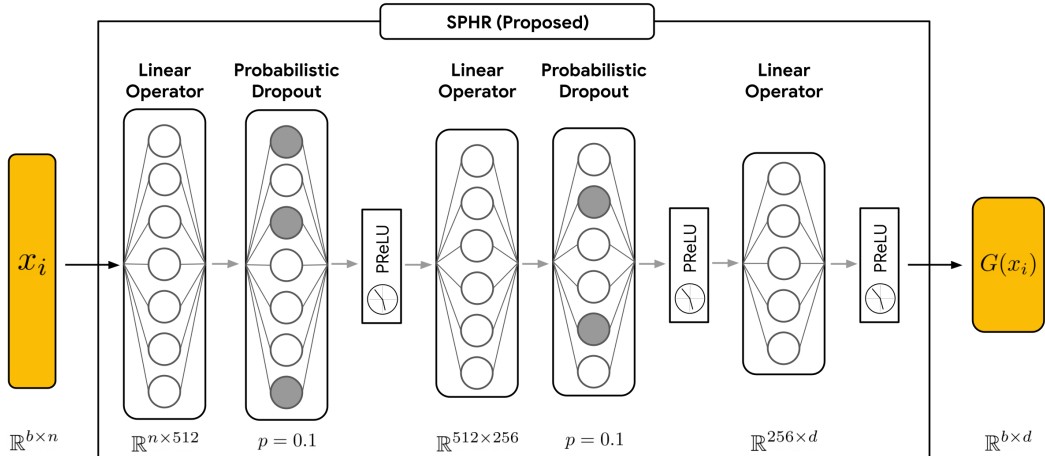

Figure A5: **Architecture of SPHR.** Our neural network is composed of three hidden layers, probablistic dropouts ($p = 0.1$) and nonlinear activations (PReLU) in between. In the figure above, $b, n$ denote to the number of patients and features, respectively, with $d$ being the output dimension (in our case $d = 32$).

For readability and reproducibility purposes, we also include a Pytorch snippet of the network used for learning representations from the UK Biobank:

```python
import torch.nn as nn

class SPHR(nn.Module):
    def __init__(self, input_dim:int = 64, output_dim:int= 32):
        self.inp_dim = input_dim
        self.out_dim = output_dim
        super().__init__()
```

Table A6: **Comparison of *multi-label* classification accuracy (*micro* F1 score) with various representations on the male patients.** We keep the same random seeds across different classifiers, and for the supervised methods, we randomly split the data into train and test (80-20) five time, and calculate the mean and standard deviation of the accuracies. Our model significantly improves the classification for all tested classifiers, demonstrating better separability in space compared to raw data and state-of-the-art method (DeepPatient).

| Model | Not-Transformed | PCA | ICA | DeepPatient | SPHR (Ours) |
|---|---|---|---|---|---|
| *KNNs* | $0.6364 \pm 0.004$ | $0.6355 \pm 0.004$ | $0.6358 \pm 0.004$ | $0.6358 \pm 0.001$ | **$0.7921 \pm 0.001$** |
| *LDA* | $0.6405 \pm 0.003$ | $0.6393 \pm 0.002$ | $0.6392 \pm 0.003$ | $0.6383 \pm 0.003$ | **$0.7811 \pm 0.002$** |
| *NN for EHR* | $0.6345 \pm 0.003$ | $0.6342 \pm 0.003$ | $0.6342 \pm 0.004$ | $0.6438 \pm 0.001$ | **$0.7930 \pm 0.003$** |
| *XGBoost* | $0.6409 \pm 0.003$ | $0.6380 \pm 0.003$ | $0.6384 \pm 0.003$ | $0.6403 \pm 0.002$ | **$0.7940 \pm 0.003$** |

Table A7: **The percentage of *apparently-healthy* male patients who develop conditions in the next immediate visit within each predicted risk group**. Among all methods (top three shown), SPHR-predicted *Normal* and *High* risk patients developed the fewest and most conditions, respectively, as expected.

| Future Diagnosis | P0 (Not-Transformed) | | | DeepPatient | | | SPHR (Ours) | | |
|---|---|---|---|---|---|---|---|---|---|
| | Normal | LR | HR | Normal | LR | Higher Risk | Normal | LR | HR |
| *Cancer* | 2.76% | 0.62% | 2.75% | 2.86% | 0.33% | <0.1% | 1.52% | 2.96% | 4.14% |
| *Diabetes* | 1.88% | 1.94% | 1.63% | 0.85% | 0.73% | 1.62% | 0.73% | 0.55% | 5.29% |
| *Other Serious Cond* | 9.57% | 6.44% | 9.47% | 9.33% | 4.41% | 7.28% | 2.73% | 8.60% | 12.45% |

```
8          self.nonlinear_net = nn.Sequential(
9                      nn.Linear(self.inp_dim,512),
10                     nn.Dropout(p=0.1),
11                     nn.PReLU(),
12                     nn.Linear(512, 256),
13                     nn.Dropout(p=0.1),
14                     nn.PReLU(),
15                     nn.Linear(256,self.out_dim),
16                     nn.PReLU()
17                     )
18
19     def forward_oneSample(self, input_tensor):
20         # useful for the forward method call and for inference
21         return self.nonlinear_net(input_tensor)
22
23     def forward(self, positive, anchor, negative):
24         ## forward method for training
25         return self.forward_oneSample(positive), self.forward_oneSample(
       anchor),  self.forward_oneSample(negative)
```

Listing 2: SPHR's network architecture

We train SPHR by minimizing our proposed NPLB objective, Eq.(3), using the Adam optimizer for 1000 epochs with $lr = 0.001$, and employ an exponential learning rate decay ($\gamma = 0.95$) to decrease the learning rate after every 50 epochs. We set the margin hyperparameter to be $\epsilon_{\circ} = 1$. In all experiments, the triplet selection was done in an offline manner using the most common triplet selection scheme (e.g. see https://www.kaggle.com/code/hirotaka0122/triplet-loss-with-pytorch?scriptVersionId=26699660&cellId=6).

## K.2  CIFAR-10 Embedding Network and Experimental Setup

To further demonstrate the improvements of NPLB on representation learning, we benchmarked various triplet losses on CIFAR10 as well. For this experiment, we trained a randomly-initialized VGG13 (Simonyan & Zisserman, 2015) model (not pre-trained) on CIFAR10 to produce embeddings $\mathbf{m} \in \mathbb{R}^{128}$ for 200 epochs, using the Adam optimizer with $lr = 0.001$ and a decaying schedule (similar to SPHR's optimization setting, as described in the main manuscript). We note that the architecture used in this experiment is identical to a traditional classification VGG model, with the difference being in the "classification" layer which is re-purposed for producing 128-dimensional embeddings. CIFAR-10 images were normalized using the standard CIFAR-10 transformation, and were not agumented during trainnig (i.e. we did not use any augmentations in training the model).

Table A8: **Studying the effects of different augmentation levels on classification as a proxy for all downstream tasks**. We followed the same procedure as all other classification experiments (including model parameters). The results for $1\times$ augmentation are omitted since the results are very similar to no augmentation.

|  | *No Augmentation* | *3×* | *5×* | *10×* |
|---|---|---|---|---|
| *Females*: Multi-Label | $0.5730 \pm 0.002$ | $\mathbf{0.6642 \pm 0.002}$ | $0.6619 \pm 0.003$ | $0.6584 \pm 0.003$ |
| *Males*: Multi-Label | $0.6247 \pm 0.005$ | $0.7845 \pm 0.003$ | $\mathbf{0.7852 \pm 0.004}$ | $0.7685 \pm 0.002$ |

### K.3 MNIST EMBEDDING NETWORK

For ease of readability and reproducibility, we provide the architecture used for MNIST as a Pytorch snippet:

```python
import torch.nn as nn

class MNIST_Network(nn.Module):
    def __init__(self, embedding_dimension=2):
        super().__init__()
        self.conv_net = nn.Sequential(
            nn.Conv2d(1, 32, 5),
            nn.PReLU(),
            nn.MaxPool2d(2, stride=2),
            nn.Dropout(0.3),
            nn.Conv2d(32, 64, 5),
            nn.PReLU(),
            nn.MaxPool2d(2, stride=2),
            nn.Dropout(0.3)
        )

        self.feedForward_net = nn.Sequential(
            nn.Linear(64*4*4, 512),
            nn.PReLU(),
            nn.Linear(512, embedding_dimension)
        )

    def forward(self, input_tensor):
        conv_output = self.conv_net(input_tensor)
        conv_output = conv_output.view(-1, 64*4*4)
        return self.feedForward_net(conv_output)
```

Listing 3: Network architecture used for validation on MNIST.

We train the network for 50 epochs using the Adam optimizer with $lr = 0.001$. We set the margin hyperparameter to be $\epsilon_\circ = 1$.

### K.4 FASHION MNIST EMBEDDING NETWORK

For readability and reproducibility purposes, we provide the architecture used for Fashion MNIST as a Pytorch snippet:

```python
import torch.nn as nn

class FMNIST_Network(nn.Module):
    def __init__(self, embedding_dimension=128):
        super().__init__()
        self.conv_net = nn.Sequential(
            nn.Conv2d(in_channels=1, out_channels=16, kernel_size=3),
            nn.PReLU(),
            nn.MaxPool2d(2, stride=2),
            nn.Dropout(0.1),
            nn.Conv2d(in_channels=16, out_channels=32, kernel_size=5),
            nn.PReLU(),
            nn.MaxPool2d(2, stride=1),
            nn.Dropout(0.2),
```

```
15              nn.Conv2d(in_channels=32, out_channels=64, kernel_size=5),
16              nn.AvgPool2d(kernel_size=1),
17              nn.PReLU()
18          )
19
20          self.feedForward_net = nn.Sequential(
21              nn.Linear(64*4*4, 512),
22              nn.PReLU(),
23              nn.Linear(512, embedding_dimension)
24          )
25
26      def forward(self, input_tensor):
27          conv_output = self.conv_net(input_tensor)
28          conv_output = conv_output.view(-1, 64*4*4)
29          return self.feedForward_net(conv_output)
```

Listing 4: Network architecture used for validation on Fashion MNIST.

We train the network for 50 epochs using the Adam optimizer with $lr = 0.001$. We set the margin hyperparameter to be $\epsilon_\circ = 1$.

### K.5 CLASSIFICATION MODELS

The parameters for *NN for EHR* were chosen based on Chen et al. (2020a). The "main" parameters for KNN and XGBoost were chosen through a randomized grid search while parameters for LDA were unchanged. We specify the parameters that were identified through grid search in the KNN and XGBoost sections.

#### K.5.1 NN FOR EHR

We follow the work of Chen et al. (2020a) and construct a feed forward neural network with the additive attention mechanism in the first layer. As in Chen *et al.* , we choose the learning rate to be $lr = 0.001$ with an L2 penalty coefficient $\lambda = 0.001$, and train the model for 100 epochs.

#### K.5.2 KNNS

We utilized the Scikit-Learn implementation of K-Nearest Neighbors. The optimal number of neighbors were found with grid search from 10 to 100 neighbors (increasing by 10). For the sake of reproducibility, we provide the parameters with scikit-learn terminology. For more information about the meaning of each parameter (and value), we refer the reviewers to the online documentation: https://scikit-learn.org/stable/modules/generated/sklearn.neighbors.KNeighborsClassifier.html.

- Algorithm: Auto
- Leaf Size: 30
- Metric: Minkowski
- Metric Params: None
- $n$ Jobs: -1
- $n$ Neighbors: 50
- $p$: 2
- Weights': Uniform

#### K.5.3 LDA

We employed the Scikit-Learn implementation of Linear Discriminant Analysis (LDA), with the default parameters.

### K.5.4 XGBOOST

We utilized the official implementation of XGBoost, located at: https://xgboost.readthed ocs.io/en/stable/. We optimized model performance through grid search for learning rate (0.01 to 0.2, increasing by 0.01), max depth (from 1 to 10, increasing by 1) and number of estimators (from 10 to 200, increasing by 10). For the sake of reproducibility, we provide the parameters the nomenclature used in the online documentation.

- Objective: Binary-Logistic
- Use Label Encoder: False
- Base Score: 0.5
- Booster: `gbtree`
- Callbacks: None,
- `colsample_by_level:1`
- `colsample_by_node:1`
- `colsample_by_tree":1`
- Early Stopping Rounds: None
- Enable Categorical: False
- Evaluation metric: None
- $\gamma$ (gamma): 0
- GPU ID: -1
- Grow Policy: depthwise'
- Importance Type: None
- Interaction Constraints: " "
- Learning Rate: 0.05
- Max Bin: 256
- Max Categorical to Onehot: 4

- Max Delta Step: 0
- Max Depth: 4
- Max Leaves: 0
- Minimum Child Weight: 1
- Missing: NaN
- Monotone Constraints: '()'
- $n$ Estimators: 50
- $n$ Jobs: -1
- Number of Parallel Trees: 1
- Predictor: Auto
- Random State: 0
- `reg_alpha:0`
- `reg_lambda:1`
- Sampling Method: Uniform
- `scale_pos_weight:1`
- Subsample: 1
- Tree Method: Exact
- Validate Parameters: 1
- Verbosity: None

---

**Algorithm 1 Proposed Augmentation of Electronic Health Records Data**. The proposed strategy will ensure that each augmented feature falls between pre-determined ranges for the appropriate gender and age group, which are crucial in diagnosing conditions.

---

**Require:** $X_{dict}$: A mapping between gender/age condition groups to raw bloodwork and lifestyle matricies

**Require:** $cond_{list}$: A list of all present conditions    # e.g. *bona fide* healthy, diabetic, etc.

**Require:** $U$: A matrix storing *upper* bounds for $feature_j$ given $condition_i$

**Require:** $L$: A matrix of the *lower* bounds for $feature_j$ given $condition_i$

1:  $\tilde{X}_{dict} \leftarrow \text{Zeros}(X_{dict})$
2:  **for** $condition_i$ in $cond_{list}$, **do**
3:      **for** $feature_j$ in $X_{dict}[condition_i]$, **do**
4:          $\mu \leftarrow \text{Mean}(feature_j)$
5:          $\sigma \leftarrow \text{STD}(feature_j)$ # Standard deviation
6:          $z \leftarrow -10^{16}$ # initialize
7:          **while** $z \notin [L_{ij}, U_{ij}]$, **do**
8:              z$\leftarrow \sim \mathcal{N}(\mu, \sigma)$ # sampled value from the Gaussian distribution
9:          **end while**
10:          $\tilde{X}_{dict}[condition_i][feature_j] \leftarrow z$ # augmented feature
11:      **end for**
12: **end for**

---

## APPENDIX L   DATA AUGMENTATION SCHEME

## APPENDIX M   : COMPLETE LIST OF FEATURES

### M.1   UKB FID TO NAME MAPPINGS FOR FEMALE PATIENTS

**Lab Metrics**
21003: Age
30160: Basophill count
30220: Basophill percentage
30150: Eosinophill count
30210: Eosinophill percentage
30030: Haematocrit percentage
30020: Haemoglobin concentration
30300: High light scatter reticulocyte count
30290: High light scatter reticulocyte percentage
30280: Immature reticulocyte fraction
30120: Lymphocyte count
30180: Lymphocyte percentage
30050: Mean corpuscular haemoglobin
30060: Mean corpuscular haemoglobin concentration
30040: Mean corpuscular volume
30100: Mean platelet (thrombocyte) volume
30260: Mean reticulocyte volume
30270: Mean sphered cell volume
30080: Platelet count

30110: Platelet distribution width
30010: Red blood cell (erythrocyte) count
30070: Red blood cell (erythrocyte) distribution width
30250: Reticulocyte count
30240: Reticulocyte percentage
30000: White blood cell (leukocyte) count
30620: Alanine aminotransferase
30600: Albumin
30610: Alkaline phosphatase
30630: Apolipoprotein A
30640: Apolipoprotein B
30650: Aspartate aminotransferase
30710: C-Reactive Protein
30680: Calcium
30690: Cholesterol
30700: Creatinine
30720: Cystatin C
30730: Gamma glutamyltransferase
30740: Glucose
30750: HbA1c
30130: Monocyte count
30190: Monocyte percentage

30760: HDL Cholesterol
30770: IGF-1
30780: LDL Direct
30810: Phosphate
30830: Sex Hormone-Binding Globulin (SHBG)
30850: Testosterone
30840: Total bilirubin
30860: Total protein
30870: Triglycerides
30880: Urate
30670: Urea
30890: Vitamin D
21001: Body Mass Index
30140: Neutrophill count
30200: Neutrophill percentage
30090: Platelet crit
**Lifestyle Metrics**
22038: MET Minutes per Week for Moderate Activity
22039: MET Minutes per Week for Vigorous Activity
22037: MET Minutes per Week for walking
22040: Summed MET Minutes per week for All Activity
22033: Summed Days of Activity
22034: Summed Minutes of Activity
1160: Sleep Duration

## M.2 UKB FID to Name Mappings for Male Patients

**Lab Metrics**
21003: Age
30160: Basophill count
30220: Basophill percentage
30120: Lymphocyte count
30180: Lymphocyte percentage
30050: Mean corpuscular haemoglobin
30060: Mean corpuscular haemoglobin concentration
30040: Mean corpuscular volume
30100: Mean platelet (thrombocyte) volume
30260: Mean reticulocyte volume
30270: Mean sphered cell volume
30080: Platelet count
30150: Eosinophill count
30210: Eosinophill percentage
30030: Haematocrit percentage
30020: Haemoglobin concentration
30300: High light scatter reticulocyte count
30290: High light scatter reticulocyte percentage
30280: Immature reticulocyte fraction

30110: Platelet distribution width
30190: Monocyte percentage
30130: Monocyte count
30750: HbA1c
30730: Gamma glutamyltransferase
30740: Glucose
30010: Red blood cell (erythrocyte) count
30070: Red blood cell (erythrocyte) distribution width
30250: Reticulocyte count
30240: Reticulocyte percentage
30000: White blood cell (leukocyte) count
30620: Alanine aminotransferase
30600: Albumin
30610: Alkaline phosphatase
30630: Apolipoprotein A
30640: Apolipoprotein B
30650: Aspartate aminotransferase
30710: C-Reactive Protein
30680: Calcium
30690: Cholesterol
30700: Creatinine
30720: Cystatin C

30760: HDL Cholesterol
30770: IGF-1
30780: LDL Direct
30810: Phosphate
30830: SHBG
30850: Testosterone
30840: Total bilirubin
30860: Total protein
30870: Triglycerides
30880: Urate
30670: Urea
30890: Vitamin D
30140: Neutrophill count
30200: Neutrophill percentage
30090: Platelet crit
**Lifestyle Metrics**
1160: Sleep Duration
21001: Body Mass Index
22038: MET Minutes per Week for Moderate Activity
22039: MET Minutes per Week for Vigorous Activity
22037: MET Minutes per Week for walking
22040: Summed MET Minutes per week for All Activity
22033: Summed Days of Activity
22034: Summed Minutes of Activity

APPENDIX N : HIGHER RESOLUTION FIGURES

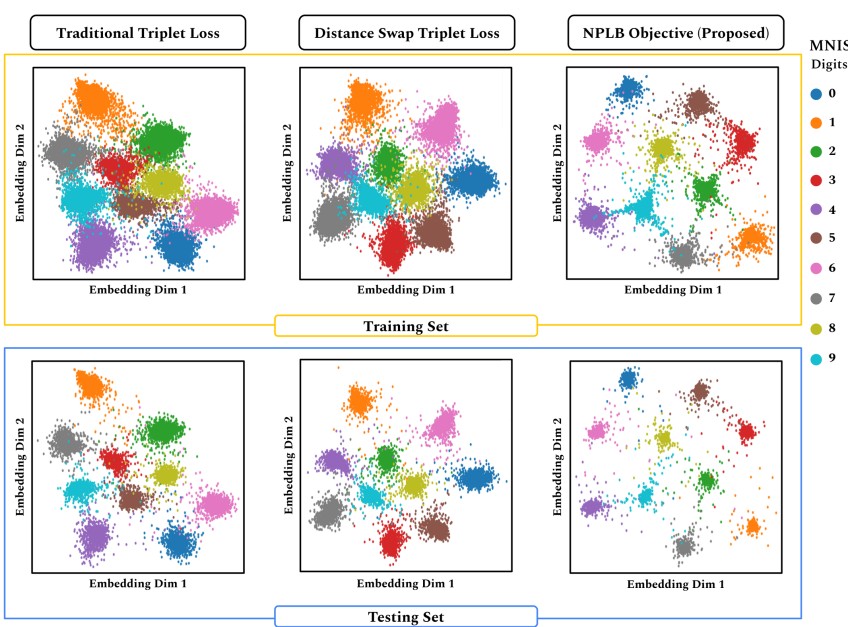

Figure A6: **Visual comparisons between traditional triplet loss (left), Distance Swap (middle) and proposed NPLB objective (right) on train (top row) and test (bottom row) sets of MNIST.** To evaluate the feasibility of proposed loss on general datasets, we trained the same network (described in Appendix Appendix K) under identical conditions on MNIST dataset, with the only difference being the loss function used. As expected by our mathematical intuition described in §3, the model trained using our proposed objective learns embeddings that are much denser within classes and farther apart from other class compared to traditional Triplet and Distance Swap variation, leading to better classification results (see Table 1).

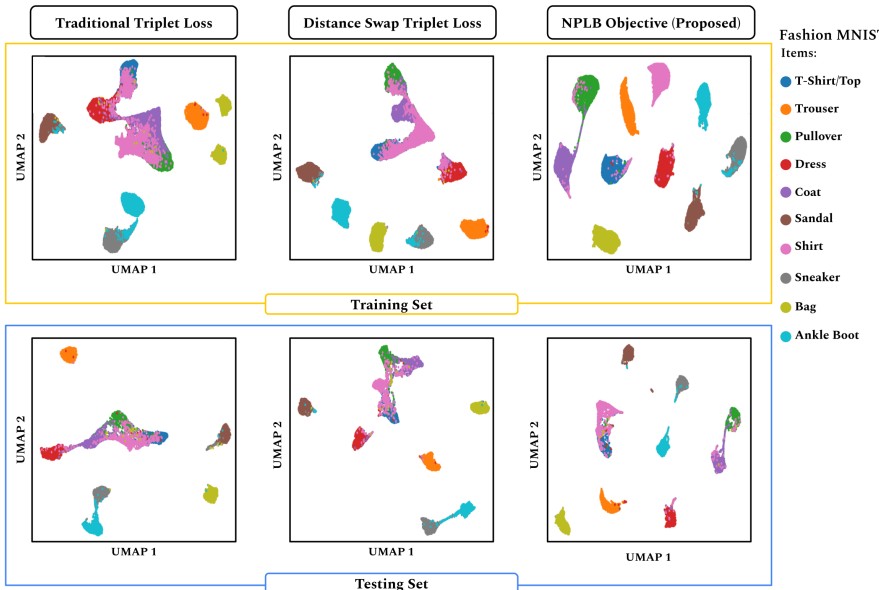

Figure A7: **UMAP-reduced comparisons between traditional triplet loss (left), Distance Swap triplet loss (middle) and proposed NPLB objective (right) on train and test sets of Fashion MNIST (top and bottom rows)**. Similar to our results on MNIST, we see both quantitative and qualitative improvements in the embeddings, as shown Table 1).

