# OpenReview forum: "No Pairs Left Behind: Improving Metric Learning with Regularized Triplet Objective"
_ICLR.cc/2023/Conference — Submitted to ICLR 2023_

### Official Review · Reviewer_7q7N · 2022-10-17

**Confidence:** 4
**Correctness:** 3
**Technical Novelty And Significance:** 1
**Empirical Novelty And Significance:** 2
**Recommendation:** 3

**Clarity, Quality, Novelty And Reproducibility:**

- The novelty of the method is very limited. Connections to other contrastive losses that make uses of more negative pairs are not explored by the authors.

- The results on MNIST and FashionMNIST should be easy to reproduce.

**Details Of Ethics Concerns:**

(A) Authors use health data and split by gender. What is the effect on other sub-groups that are not split?

(B) Could enforcing equidistant negatives results in fairness concerns related to common fairness metrics? Unfortunately I am unfamiliar with the field.

**Strength And Weaknesses:**

Strengths
- The authors provide results for interesting applications of the method on healthcare data.

Weaknesses

- The proposed idea is extremely simple. In essence, the submission uses the fact that a triplet contains two negative pairs: (a,n) and (p,n). In addition to the standard triplet loss, the authors add a loss that encourages the distance between the p and n pair (a negative pair) to be similar to the distance between the a and the n pair (another negative pair). Thus, the paper just adds an extra loss to the typical triplet loss that encourages all negative pairs in the dataset to be roughly equidistant.

- Given the extreme simplicity of this idea, in my opinion the paper is lacking: (A) a thorough analysis how it compares to other recent triplet/contrastive losses, and in what cases equidistant negative pairs are desirable, (B) a thorough benchmarking comparing to more related work on triplet and contrastive losses*, and (C) an evaluation on a much larger set of common (metric learning) benchmark datasets used in related contrastive/triplet methods, rather than just MNIST and FashionMNIST.

      *To just name a few methods for which I think comparisons are lacking: Lifted Structured Loss [1], Multi-Class N-pair loss [2], Noise Contrastive Estimation [3], InfoNCE [4], Circle Loss [5]

      [1] Oh Song, H., Xiang, Y., Jegelka, S., & Savarese, S. (2016). Deep metric learning via lifted structured feature embedding. In Proceedings of the IEEE conference on computer vision and pattern recognition (pp. 4004-4012).

      [2] Sohn, K. (2016). Improved deep metric learning with multi-class n-pair loss objective. Advances in neural information processing systems, 29.

      [3] Gutmann, M., & Hyvärinen, A. (2010, March). Noise-contrastive estimation: A new estimation principle for unnormalized statistical models. In Proceedings of the thirteenth international conference on artificial intelligence and statistics (pp. 297-304). JMLR Workshop and Conference Proceedings.

      [4] Oord, A. V. D., Li, Y., & Vinyals, O. (2018). Representation learning with contrastive predictive coding. arXiv preprint arXiv:1807.03748.

      [5] Sun, Yifan, et al. "Circle loss: A unified perspective of pair similarity optimization." Proceedings of the IEEE/CVF Conference on Computer Vision and Pattern Recognition. 2020.



- Finally, the motivation via the definition of uniform in-class embeddings is unclear to me, in part because the notation is not well defined at the moment. I encourage the authors to elaborate more on this point. Notation issue: Rojas-Thomas & Santos (2021)  define the local density (LD) over a point x_i in c_k. The authors in this submission define LD on c_k directly, without specifying if p_j is also in c_k or any data point in the dataset.



**Summary Of The Paper:**

The authors introduce a simple modification to a metric-learning triplet loss by adding penalty term which encourages the distance between the anchor sample and negative  sample to be similar to the distance between the positive sample and the negative sample.



**Summary Of The Review:**

I vote reject as I do not think that this work is ready for publication, given that the comparison to related triplet/contrastive methods on public benchmark datasets is severely limited. Adding more related methods and experiments should not present a significant hurdle.


--------
## Post Rebuttal Update
I thank the authors for working to improve their submission. I am keeping my score since metric learning work published at top ML conferences should follow a more thorough benchmarking on diverse datasets and against recent competitive metric learning approaches. In addition, in its current form, the submission is still lacking convincing arguments as to why the proposed approach should work better than alternative metric learning approaches that also make use of additional relationships between positive and/or negative pairs. I encourage the authors to take this feedback into account to mature the submission.

---

> ### Author Response · Authors · 2022-11-18
> **Response to Reviewer 7q7N comments**
>
> > Given the extreme simplicity of this idea, in my opinion the paper is lacking:
> A thorough benchmarking comparing to more related work on triplet and contrastive losses*:
>
> 1. *[1] Oh Song, H., Xiang, Y., Jegelka, S.,  Savarese, S. (2016). Deep metric learning via lifted structured
> feature embedding.  In Proceedings of the IEEE conference on computer vision and pattern recognition (pp. 4004-4012).*
>
> 1. *[2] Sohn, K. (2016). Improved deep metric learning with multi-class n-pair loss objective. Advances
> in neural information processing systems, 29.*
>
> 1. *[4] Oord, A. V. D., Li, Y.,   Vinyals, O. (2018).  Representation learning with contrastive predictive
> coding. arXiv preprint arXiv:1807.03748.*
>
> * We agree with the reviewer that the comparison of these methods against our proposed formulation
> was missing from the manuscript.  We now have added a brief description of these works as well as
> the benchmark comparisons (shown in Tables 1 and 3 with a complete version presented as Table A2
> of the Appendix) to our revised manuscript.
>
> > [3] Gutmann, M.,  Hyvärinen, A. (2010, March). Noise-contrastive estimation: A new estimation prin-
> ciple for unnormalized statistical models. In Proceedings of the thirteenth international conference on
> artificial intelligence and statistics (pp. 297-304). JMLR Workshop and Conference Proceedings.
>
> * We thank the reviewer for suggesting these valuable references.  While we agree that this approaches
> is related, we did not include this methods in our analyses given limitations on response time and the
> scope of our manuscript.
>
> > An evaluation on a much larger set of common (metric learning) benchmark datasets used in related contrastive/triplet methods, rather than just MNIST and FashionMNIST.
>
> - We appreciate the reviewer’ suggestion on additional benchmarking datasets. To address this comment and
> per suggestion of Reviewer LF9L, we have added benchmarking results on CIFAR10 as well. Additionally
> in our online code repository, we will include tutorial notebooks for testing our approach on two other
> datasets as well (namely CIFAR100 and MIMIC).
>
> > Finally, the motivation via the definition of uniform in-class embeddings is unclear to me, in part because the notation is not well defined at the moment. I encourage the authors to elaborate more on this point. Notation issue: Rojas-Thomas \& Santos (2021) define the local density (LD) over a point x_i in c_k. The authors in this submission define LD on c_k directly, without specifying if p_j is also in c_k or any data point in the dataset.
>
> * We thank the reviewer for the clarifying question. We completely agree with the reviewer that the current notation is not clear for the reader. Per the suggestion of the reviewer, we have reverted to using the same notation as in Rojas-Thomas \& Santos (2021), changing the related text to the following:
>
> *Given a metric learning model $\phi$, let local density of a sample $p_i$ be defined as $LD(p_i)_{p_i\in c_k} = \min\{d(\phi(p_i),\phi(p_j))\}$, $\forall p_i \in c_k$ and $i\neq j$, and let $AD(c_k)$ be the average local density of all point in class $c_k$.*

---

> > ### Comment · Reviewer_7q7N · 2022-11-18
> > **Response to the authors' comments**
> >
> >  I'd like to commend the authors for working to improve their manuscript. I still have a number of concerns and I do not think this submission is ready for publication at ICLR at this time.
> >
> > - Missing: Analysis why it works or should work. The authors fixed the notation of the motivating related work statement. However, the authors make multiple leaps in their argument as to why the in-class uniformity is inversely related to the equidistant negative statement (|d(p,n) -d(a,n)|). To me, at least in its current form, the argument is neither convincing nor complete.
> > - Datasets: All benchmark datasets have even class balance and 10 classes. I can see why having equidistant negatives can be beneficial here. I believe that a mature submission to ICLR on metric learning should benchmark datasets with more diversity (in terms of number of classes, data type, difficulty, etc).
> > - The results themselves
> >     - CIFAR10: related metric learning work [6] reports better performance on CIFAR10 with a resnet18, and a 3 NN classifier, even when only Euclidean distance is used (although the classifier is pretrained). The fact that the reported performance here is lower than this result using Euclidean distance makes me hesitant to read too much into the reported results. The authors of the present submission use  80% of the CIFAR training data to train the VGG13 model (with ground truth encapsulated in triplets). To me, the bar should be to at least beat a Euclidean distance transfer learning result. I don't want to read too much into it, but it underscores why I believe that the experiments are not sufficient overall in coverage of related work + breadth of datasets.
> > - Minor: REPRODUCIBILITY  paragraph is beyond page 9. Should be removed or included in main text up to page 9.
> >
> > [6] Qian, Q., Tang, J., Li, H., Zhu, S., & Jin, R. (2018). Large-scale distance metric learning with uncertainty. In Proceedings of the IEEE Conference on Computer Vision and Pattern Recognition (pp. 8542-8550).

---

### Official Review · Reviewer_wL6B · 2022-10-25

**Confidence:** 3
**Correctness:** 3
**Technical Novelty And Significance:** 2
**Empirical Novelty And Significance:** 2
**Recommendation:** 5

**Clarity, Quality, Novelty And Reproducibility:**

The paper seems to contribute incrementally to the spectrum of triplet loss objectives. The empirical evaluation setting could have been explained a bit more clearly. For example, it is not clear if all the test subjects are covered by classes “Normal”, “Low Risk” and “High Risk” in section 5.3? Why is the dataset split based on the gender and presented separately for female (and male in appendix)? Also, why is DeepPatient transformation R^n -> R^n and not R^n -> R^d like others? Given that repository will be shared, the results seem reproducible.

**Strength And Weaknesses:**

Strengths
Motivation for, and derivation of the new triplet objective is clear.
Empirical results for cluster density and separability, as well as embeddings utility in downstream prediction tasks looks compelling

Weaknesses
At moments it is not easy to follow the flow of the paper. Last paragraph of the Introduction appears to have redundancy in summarizing the main 3 aspects of the contribution: novel variation of triplet objective, embeddings’ utility for classification tasks, and distance based risk indicators. Claim that this approach does not require additional hyperparameters, as it is based on distances, might be misleading. In equation 3, the added “regularization term”, ie the square of distance, could have a hyperparameter multiplicator to tune the impact on overall loss. Just in this case it is set to 1 (similarly as it is set to 0.1 in MDR case).

Minor suggestions
Equation 1 has an extra right bracket.


**Summary Of The Paper:**

A variation of a triplet loss for deep metric learning, dubbed NPLB, is proposed. It is inspired by the condition that distance between positive and negative examples should be bigger than distance between anchor and positive. It is illustrated that optimizing such objective leads to more compact clusters. Empirical results on benchmark datasets (MNIST and fashionMNIST) and Biobank dataset show increased prediction accuracy (weighted F1 score), when using embeddings from NPLB.

**Summary Of The Review:**

Given that modeling novelty seem to be adding distance based regularization to known objective, and novel application of using the learned metrics as health risk is not quite clear to me, I am leaning towards the rejection, although open to turn to the positive side if my concerns are addressed.

---

> ### Author Response · Authors · 2022-11-18
> **Response to Reviewer wL6B**
>
> > 1.  Weaknesses At moments it is not easy to follow the flow of the paper. Last paragraph of the Introduction appears to have redundancy in summarizing the main 3 aspects of the contribution: novel variation of triplet objective, embeddings’ utility for classification tasks, and distance based risk indicators.
>
> * We thank the reviewer for their valuable feedback. We have clarified our notation and rewritten sections of the original manuscript to help with the flow of the paper.
>
> > 2.  Claim that this approach does not require additional hyperparameters, as it is based on distances, might be misleading. In equation 3, the added “regularization term”, ie the square of distance, could have a hyperparameter multiplicator to tune the impact on overall loss. Just in this case it is set to 1 (similarly as it is set to 0.1 in MDR case).
>
> * We appreciate the reviewer raising this point, and we agree that the language may be misleading due to the points raised by the reviewer. We have now changed the sentence to read as:
> A drawback of regularization methods is the choice of hyperparameter that balances the terms in the loss, though adaptive balancing methods could be used (Chen et al., 2018; Heydari et al., 2019).
>
> > 3.  Minor suggestions Equation 1 has an extra right bracket.
>
> * We thank the reviewer for catching our mistake. Eq. (1) is fixed in the revised manuscript.
>
> > 4.  The empirical evaluation setting could have been explained a bit more clearly. For example, it is not clear if all the test subjects are covered by classes “Normal”, “Low Risk” and “High Risk” in section 5.3?
>
> * Our risk predictions will assign each patient to one of the pre-defined risk groups (potentially more granular risk groups if desired). In our case, this means that all patients’ will be mapped to the "Normal", "Low Risk" or "High Risk" groups. We have rewritten the related part of the manuscript to make this point more explicit.
>
> > 5.  Why is the dataset split based on gender and presented separately for female (and male in appendix)?
>
> * We thank the reviewer for this great question.  Recent works in the literature (e.g.  Cohen et al., 2021)  have shown a significant variability between male and female patients.  Additionally, the UK Bank includes certain sex-specific (e.g.  medications) that we wanted to use in our modeling.  As a result, we chose to split the subjects based on biological sex to ensure the completeness of features and to keep our findings clinically relevant.
>
> > 6.  Why is DeepPatient transformation R^n -> R^n and not R^n -> R^d like others?
>
>
> * We thank the reviewer for pointing out the error in our writing. The original latent representation of deep patients is a mapping from $\mathbb{R}^n \to \mathbb{R}^m$ (with $m=500$ in the Miotto et al. implementation). To address the reviewer's comment, we added an extra layer in the encoder of DeepPatient to map $\mathbb{R}^m \to \mathbb{R}^d$ now, making the encoder an operator going from $\mathbb{R}^n \to \mathbb{R}^d$, as suggested by the reviewer. We have updated the DeepPatient results in our manuscript to reflect this change.

---

> > ### Comment · Reviewer_wL6B · 2022-11-18
> > **Predicting health risks from a single lab visit**
> >
> > Thanks to the authors for clarifying several points. I have an additional comment. Maybe results in a Table 4 could be presented in another way. From the current presentation, where for Deep Patient in LR and HR groups there are very few patients who develop a condition <0.1%, it is not clear if DeepPatient assigned most of patients into Normal group. Or if the groups were of appropriate sizes, but the DeepPatient missed to put the ones with higher risk in LR and HR groups.

---

> > > ### Author Response · Authors · 2022-11-19
> > > **Clarifying Single-Time Health Risk Predictions for Healthy Patients**
> > >
> > > > *Thanks to the authors for clarifying several points. I have an additional comment. Maybe results in a Table 4 could be presented in another way. From the current presentation, where for Deep Patient in LR and HR groups there are very few patients who develop a condition <0.1%, it is not clear if DeepPatient assigned most of patients into Normal group. Or if the groups were of appropriate sizes, but the DeepPatient missed to put the ones with higher risk in LR and HR groups.*
> > >
> > > Thank you very much for your clarifying question. To address your comment, we would like to clarify our future health risk prediction experiment here (we have updated Table 4 of the manuscript as well): Using the embeddings from each method and our definition of health risk, we partition a fixed number of *currently healthy* subjects into one of the three risk groups (“Normal”, “Lower Risk” and “Higher Risk”). The proportion of subjects that fall into each of these three risk groups has now been added Table 4 of the revised manuscript (e.g. using DeepPatient embeddings, roughly 77% of subjects are classified as Normal, 8% as Lower Risk and 15% as Higher Risk).
> > >
> > > To validate our single-time predictions, we consider the *next* assessment of the subjects and calculate the percentage of those who developed health conditions within each risk group. These are the results that were originally presented in Table 4. Per your suggestion, we have now updated Table 4 to include both the percentage of subjects in each health risk, as well as the percentage of those who develop conditions in the future within each risk group. We have also updated the caption of Table 4 to better clarify the points discussed here.
> > >
> > > We hope that our explanation here and the updated Table 4 in the revised manuscript have addressed your comment. Please let us know if you have any additional questions or concerns.

---

### Official Review · Reviewer_LF9L · 2022-10-25

**Confidence:** 4
**Correctness:** 3
**Technical Novelty And Significance:** 2
**Empirical Novelty And Significance:** 2
**Recommendation:** 3

**Clarity, Quality, Novelty And Reproducibility:**

- How stable is the model? In table 5, it would be more meaningful to also re-train the embeddings.
- It is unclear whether the very small improvements in a toy-like dataset such as MNIST are meaningful. It would be more insightful to see results on a slightly more complex dataset such as CIFAR-10 or CIFAR-100.
- How does the proposed approach scale?
- How does the proposed approach work for EHR data in an unsupervised setting (online generation of triplets via negative sampling)? I don't really see the practical relevance of the supervised setting here and importantly comparisons to unsupervised methods PCA and ICA are not meaningful.
- PCA and ICA are very poor baselines as dimensionality reduction methods. It would be more meaningful to see results based on eg MDS, a VAE, kernel PCA/GP-LVM, diffusion maps,...
- In Fig 3, how was the UMAP for non-transformed data computed? Based on the PCA representation?

**Strength And Weaknesses:**

While the approach is interesting, I have major concerns regarding the empirical analysis.



**Summary Of The Paper:**

The authors propose a novel formulation of the triplet objective function by explicitly regularizing the distance between the positive and negative samples in a triplet.
They evaluate their approach on (Fahsion) MNIST and EHR data.



**Summary Of The Review:**

Major flaws in empirical analysis.

---

> ### Author Response · Authors · 2022-11-18
> **Response to Reviewer LF9L's comments**
>
> > 1.  How stable is the model? In Table 5, it would be more meaningful to also re-train the embeddings.
>
> * We analyzed the stability of our proposed deep learning model for EHR data (SPHR) as follows:  We addedGaussian noise (due to the preprocessing, we chose a distribution centered at zero and 0.1 standard deviations)to all features at random for 5%, 10%, 25%, and 50% of subjects at random.  We found that our results were comparable to the case with no noise injection, with the worst case being  3% drop in the weighted F1 score for classification when 50% of subjects were perturbed.  Unfortunately, we are unsure about the comment by the reviewer about retraining embeddings in table 5, given that our manuscript does not have Table 5.  We would appreciate clarification on this comment by the reviewer, if possible, so that we could address the second part of this comment as well.
>
> > 2.  It is unclear whether the very small improvements in a toy-like dataset such as MNIST are meaningful. It would be more insightful to see results on a slightly more complex dataset such as CIFAR-10 or CIFAR-100.
>
> * We appreciate the suggestion of applying our method to more complex datasets, such as CIFAR-10.  We have added a new table in the row of Table 1, which shows consistent results with other tested datasets:  We trained a VGG13 model (with the last "classification" layer re-purposed for producing embeddings) using the variousTriplet objectives,  including the proposed NPLB. We found that,  on average,  our approach outperforms the second-best method by more than 2% (Weighted F1 Score) and the traditional triplet formulation by more than4%.
>
> > 3.  How does the proposed approach scale?
>
> * Given that our objective function does not introduce additional computational cost, we measured the scalability of the deep EHR embedding model proposed, SPHR, using different numbers of training samples.  We have added the scalability results for SPHR in Table A1 of the Appendix, showing the triplet mining time and training duration for varying number of triplets on CPUs and GPUs.
>
> > 4.  How does the proposed approach work for EHR data in an unsupervised setting (online generation of triplets via negative sampling)? I don’t really see the practical relevance of the supervised setting here and importantly comparisons to unsupervised methods PCA and ICA are not meaningful.
>
> * We agree with the reviewer that simple supervised classification problems may be of limited practical significance to EHR studies.  However, we note that the primary goal of our approach is to predict future health risks for the current healthy population using a single time point. In this setting, it is expected to have defined labels(i.e.  diagnosed conditions or lack of) for patients which could be leveraged for learning meaningful representations that could be used for other downstream tasks.  In this setting, our approach can easily be extended to online triplet mining strategies, such as "Batch All" strategy (where a pairwise distance between all samples in the batch are calculated and the non-valid triplets are masked) or "hard-negative/hard-positive" mining.
>
> > 5.  PCA and ICA are very poor baselines as dimensionality reduction methods. It would be more meaningful to see results based on eg MDS, a VAE, kernel PCA/GP-LVM, diffusion maps, etc.
>
> * We completely agree with the reviewer that PCA and ICA are too simplistic for representing complex datasets,such as electronic health records. However, due to their simplicity, most health informaticians are most familiar with PCA and ICA, and are therefore used by many for representing healthcare data and benchmarking (Riccardo Miotto et al., 2016; Zhongheng Zhang et al, 2017; Sheikh S. Abdullah et al., 2020). Our goal in including these methods for our analysis was to (1) provide a comparison against other existing methods that use linear dimensionality reduction methods, and (2) to argue that linear methods (such as PCA and ICA) are often not suitable for representing EHR data, as also suggested by the reviewer.To address the reviewer’s comments, we have now replaced ICA column of Tables 2 of the main manuscript with Diffusion Maps (a non-linear transformation). Our findings show an improvement over the linear methods(PCA and ICA) but not as well as the deep-learned methods (shown in Table 2).
>
> > 6.  In Fig 3, how was the UMAP for non-transformed data computed? Based on the PCA representation?
>
> * We thank the reviewer for the clarifying question.  Although many approaches utilize the first n-many PCs as the data representation to perform t-sne or UMAP, we chose to perform UMAP on the original representations given the dimensionality of the raw/transformed data. It is important to note that we also tried PCA+UMAP (30PCs) on the raw EHRs, but the visualization seemed very similar to the UMAP on the original feature space.

---

### Author Response · Authors · 2022-11-18
**General Response to All Reviewers and Summary of Major Updates**

We would like to thank the reviewers for taking the time to carefully assess our manuscript and provide valuable comments and suggestions. We appreciate the opportunity to improve our work and provide the revised manuscript for further consideration. In the revised main manuscript and appendix, we addressed most of the concerns raised by the reviewers. In doing so, we believe that our manuscript is significantly improved.

We are glad that the reviewers found our approach interesting (*Reviewer LF9L*), the motivation and derivation of our new triplet objective clear (Reviewer wL6b), and our results and application to healthcare interesting (*Reviewers 7q7N* and *wL6b*). To address concerns raised by the reviewers, we have made the following changes in our work and noted them in each individual response when possible. The major changes are the following:

* Added additional benchmarking (on CIFAR 10), as suggested by *Reviewer LF9L* and *Reviewer 7q7N*, with the results added to Tables 1 and 2A.

* Compared our method with three additional state-of-the-art contrastive approaches, as suggested by *Reviewer 7q7N*, and added the results to Tables 1, 3 and 2A.


* Performed stability analysis of the proposed deep-learned embeddings to address *Reviewer LF9L*'s question.

* Replaced a linear transformation method with a non-linear approach for benchmarking performance, as suggested by *Reviewer LF9L*. The results are presented in Table 2.

* Conducted a large-scale scalability analysis of our proposed deep learning model, as asked by *Reviewer LF9L*.

* Clarified the shortcomings of regularization-based methods, namely the balancing hyperparameters, as suggested by *Reviewer wL6B*. Additionally, we added references for adaptive weighting of multi-task objective functions.

* Edited the main manuscript to clarify our writing and notations as suggested by *Reviewers wL6B* and *7q7N*.


We hope that our additional experiments and responses have adequately addressed all of the reviewers' concerns. We would like to thank all reviewers again for their time and feedback, and would truly appreciate it if the reviewers could re-evaluate their initial scores after assessing our responses and updated work.

---

### Decision · Program_Chairs · 2023-01-20

**Decision:**

Reject

**Justification For Why Not Higher Score:**

Paper not ready for publication at ICLR due to limited novelty and too narrow scope of empirical evaluation.

**Justification For Why Not Lower Score:**

n/a

**Metareview: Summary, Strengths And Weaknesses:**

This submission has been thoroughly reviewed by three knowledgeable reviewers. All of them assessed it below the threshold for acceptance. The reviewers brought up several concerns, including incrementality of the contribution to the field, and concerns with the scope and soundness of the empirical evaluation framework. This paper is unfortunately not ready for publication at ICLR at this point.